# On the Hardness of Approximating Distributions with Tractable Probabilistic Models

**John Leland**    **YooJung Choi**
School of Computing and Augmented Intelligence
Arizona State University
jslelan1@asu.edu, yj.choi@asu.edu

## Abstract

A fundamental challenge in probabilistic modeling is to balance expressivity and inference efficiency. Tractable probabilistic models (TPMs) aim to directly address this tradeoff by imposing constraints that guarantee efficient inference of certain queries while maintaining expressivity. In particular, probabilistic circuits (PCs) provide a unifying framework for many TPMs, by characterizing families of models as circuits satisfying different structural properties. Because the complexity of inference on PCs is a function of the circuit size, understanding the size requirements of different families of PCs is fundamental in mapping the trade-off between tractability and expressive efficiency. However, the study of expressive efficiency of circuits are often concerned with exact representations, which may not align with model learning, where we look to approximate the underlying data distribution closely by some distance measure. Moreover, due to hardness of inference tasks, exactly representing distributions while supporting tractable inference often incurs exponential size blow-ups. In this paper, we consider a natural, yet so far under-explored, question: *can we avoid such size blow-up by allowing for some small approximation error?* We study approximating distributions with probabilistic circuits with guarantees based on $f$-divergences, and analyze which inference queries remain well-approximated under this framework. We show that approximating an arbitrary distribution with bounded $f$-divergence is NP-hard for any model that can tractably compute marginals. In addition, we prove an exponential size gap for approximation between the class of decomposable PCs and that of decomposable and deterministic PCs.

## 1   Introduction

The expressive power of probabilistic and generative models has increased rapidly in recent years: from Bayesian networks [18], GANs [26], VAEs [29], and normalizing flows [35], to diffusion models [27] and transformers [46] have demonstrated remarkable success in capturing complex distributions. Yet, despite their expressivity, many of these models do not support efficient computation of fundamental probabilistic queries, such as marginals and conditionals, which are critical for inference in domains such as healthcare [40], neuro-symbolic AI [43], environmental science [5], and algorithmic fairness [9].

Tractable probabilistic models—such as probabilistic circuits [8], probabilistic generating circuits [50], determinantal point processes [1], and more—address the need for probabilistic queries by balancing expressivity and tractable inference, achieved through enforcing constraints on the models. To more compactly describe the constraints enforced, we focus on probabilistic circuits, which also have extensive literature on the conditions of tractability [8]. While these structural constraints enable efficient inference, they introduce a tradeoff by potentially affecting the models' ability to compactly represent distributions (i.e., their expressive power). Naturally, there have been many

39th Conference on Neural Information Processing Systems (NeurIPS 2025).

works characterizing the expressive efficiency of different circuit classes [19, 2, 8, 48, 49, 20]—i.e., their ability to compactly and *exactly* represent certain classes of functions or distributions. However, the approximate case—how structural constraints affect the ability to *approximately represent* distributions, remains comparatively underexplored. This motivates a shift in focus from exact to approximate modeling: representing a distribution approximately within a small distance under some metric.

This shift raises a fundamental question: *does allowing a small approximation error alleviate the exponential separation between circuit classes observed in exact modeling, or does hardness persist even in the approximate setting?* Our motivation for this study is two-fold. (1) In learning probabilistic circuits from finite data, often the goal is not necessarily to exactly represent some known distribution but rather to approximate it as closely as possible with a PC of reasonable size. Thus, showing that certain distributions cannot be approximated within a bounded distance by a compact PC satisfying some structural properties implies that any learning algorithm whose hypothesis space is that family of PCs would fail to learn the distribution with a bounded approximation error. (2) Moreover, probabilistic circuits can also be used to perform inference on other probabilistic models (such as Bayesian networks or probabilistic programs) by compiling them into PCs then running efficient inference on the compiled circuits [4, 28, 16, 7, 25]. This suggests the following approximate inference scheme: *approximately* compile a probabilistic model into a PC then run efficient *exact* inference on the approximately compiled PC. Moreover, if we could bound the distance between the target distribution and approximate model, we can hope to provide guarantees on the approximate inference results as well.

Our main contributions are as follows: (1) we prove that it is NP-hard to approximate distributions within a bounded $f$-divergence using *any model that supports tractable marginals*, with proof via a reduction from SAT (Theorem 3.4 and 3.5); (2) we derive an unconditional, exponential separation between decomposable PCs and decomposable & deterministic PCs for approximate modeling (Theorem 4.1); (3) we study the relationship between bounds on divergence measures for approximate modeling and approximation errors for marginal and maximum-a-posteriori (MAP) inference, characterizing when one is or is not sufficient to guarantee the other (Sections 3.1 and 5).

## 2 Preliminaries

**Notations**   We use uppercase letters $(X)$ to denote random variables and lowercase letters $(x)$ to denote assignments to these random variables. Sets of random variables and assignments are denoted using bold letters ($\mathbf{X}$ and $\mathbf{x}$). The *accepting models* (i.e., satisfying assignments) of a Boolean function $f : \{0,1\}^n \to \{0,1\}$ over $n$ variables is denoted by $f^{-1}(1)$. The number of accepting models of $f$ is referred to as $\mathrm{MC}(f)$, a shorthand for its *model count*. Moreover, a Boolean function $f$ is the *support* of a distribution $P$, if $P$ is non-zero only over the models of $f$.

### 2.1 Probabilistic Circuits

Probabilistic circuits (PCs) [8] provide a unifying framework for a wide class of tractable probabilistic models, including arithmetic circuits [15], sum-product Networks [37], cutset networks [38], probabilistic sentential decision diagrams [30], and bounded-treewidth graphical models [24, 11].

**Definition 2.1** (Probabilistic circuits)**.**   A probabilistic circuit (PC) $\mathcal{C} := (\mathcal{G}, \theta)$ represents a joint probability distribution $p(\mathbf{X})$ over random variables $\mathbf{X}$ through a directed acyclic graph (DAG) $\mathcal{G}$ parameterized by $\theta$. The DAG is composed of 3 types of nodes: leaf, product $\otimes$, and sum $\oplus$ nodes. Every leaf node in $\mathcal{G}$ is an input, and every internal node receives inputs from its children $\mathrm{in}(n)$. The scope of a given node, $\phi(n)$, is a recursively defined function which associates to each unit $n$ a subset of $\mathbf{X}$: for each non-input unit $n$, $\phi(n) = \cup_{c \in \mathrm{in}(n)} \phi(c)$, and the scope of a leaf node is a single variable in $\mathbf{X}$. Naturally, the scope of the root node is $\mathbf{X}$. Each node $n$ of a PC is then recursively defined as:

$$p_n(\mathbf{x}) := \begin{cases} l(x), & \text{if } n \text{ is a leaf} \\ \prod_{c \in \mathrm{in}(n)} p_c(\mathbf{x}) & \text{if } n \text{ is a } \otimes \\ \sum_{c \in \mathrm{in}(n)} \theta_{n,c} p_c(\mathbf{x}) & \text{if } n \text{ is a } \oplus \end{cases} \tag{1}$$

where $\theta_{n,c} \in [0,1]$ is the parameter associated with the edge connecting nodes $n, c$ in $\mathcal{G}$, and $\sum_{c \in \text{in}(n)} \theta_{n,c} = 1$. In this paper, we assume $l(x)$ at a leaf node is a Boolean indicator function: i.e., $\mathbb{1}[x = 1]$ or $\mathbb{1}[x = 0]$. The distribution represented by the circuit is the output at its root node.

A key characteristic of PCs is that imposing certain structural properties on the circuit enables tractable (polytime) computation of various queries. In this paper we focus on two families of PCs: those that are tractable for *marginal* inference and for *maximum-a-posteriori (MAP)* inference.

The class of marginal queries of a joint distribution $p(\mathbf{X})$ over variables $\mathbf{X}$ refers the set of functions that can compute $p(\mathbf{y})$ for some assignment $\mathbf{y}$ for $\mathbf{Y} \subseteq \mathbf{X}$. Marginalization is a fundamental statistical operation which enables reasoning about subsets of variables, essential for tasks such as decision making, learning, and predicting under uncertainty. While marginal inference is #P-hard in general [39, 13], the family of PCs satisfying the following structural conditions admit tractable marginal inference—specifically in linear time in the size of the circuit [17].

**Definition 2.2** (Smoothness and decomposability). A sum unit is *smooth* if its children have identical scopes: $\phi(c) = \phi(n), \forall c \in \text{in}(n)$. A product unit is *decomposable* if its children have disjoint scopes: $\phi(c_i) \cap \phi(c_j) = \emptyset, \forall c_i \neq c_j \in \text{in}(n)$. A PC is smooth and decomposable iff every sum unit is smooth and every product unit is decomposable.

In addition, we are often interested in finding the most likely assignments given some observations. The class of maximum-a-posteriori (MAP)[1] queries of a joint distribution $p(\mathbf{X})$ is the set of queries that compute $\max_{\mathbf{q} \in val(\mathbf{Q})} p(\mathbf{q}, \mathbf{e})$ where $\mathbf{e} \in val(\mathbf{E})$ is an assignment to some subset $\mathbf{E} \subseteq \mathbf{X}$ and $\mathbf{Q} = \mathbf{X} \backslash \mathbf{E}$. Again, MAP inference is NP-hard in general [42] but can be performed tractably for a certain class of PCs. In particular, smoothness and decomposability are no longer sufficient, and we must enforce an additional condition.

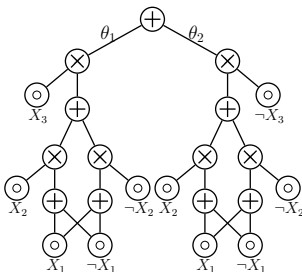

**Definition 2.3** (Determinism). A sum node is *deterministic* if, for any fully-instantiated input, the output of at most one of its children is nonzero. In other words, the supports of its children are mutually disjoint. A PC is deterministic iff all of its sum nodes are deterministic.

Figure 1: A smooth, decomposable, and deterministic PC (weights shown only for the root for conciseness).

Figure 1 depicts an example PC that is smooth, decomposable, and deterministic, which thus supports tractable marginal as well as MAP inference.[2]

**Logical circuits** Probabilistic circuits are closely related to *logical circuits* in the *knowledge compilation* literature [19]. Logical circuits encode Boolean functions as directed acyclic graphs consisting of AND ($\wedge$) and OR ($\vee$) gates with positive and negative literals as leaf nodes. We can also characterize different families of logical circuits based on their structural properties: e.g., decomposable negation normal forms (DNNFs) and deterministic decomposable negation normal forms (d-DNNFs).[3] There is a rich literature studying different logical circuit families in terms of their tractability for inference and operations, as well as their relative *succinctness (expressive efficiency)* for both exact and approximate compilation, which we will leverage for our hardness results and size lower bounds on probabilistic circuits.

## 2.2 Measures of Difference between Probability Distributions

To study the hardness of approximating probability distributions, we first need to be able to measure how "good" an approximation is. In particular, we focus on the class of $f$-divergences.

---

[1] Sometimes also called the most probable explanation (MPE).

[2] For PCs over Boolean variables, a weaker form of decomposability called *consistency* [37, 8] actually suffices instead of decomposability for both tractable marginal and MAP inference. In this paper, we still focus on classes of PCs that are decomposable as they are the most commonly considered, both as learning targets as well as for characterizing expressive efficiency.

[3] Structural conditions are same as before (Definitions 2.2 and 2.3), except that smoothness and determinism apply to OR gates and decomposability to AND gates.

**Definition 2.4** ($f$-divergence [36]). Let $f : (0, \infty) \to \mathbb{R}$ be a convex function with $f(1) = 0$, and $P, Q$ be two probability distributions over a set of Boolean variables $\mathbf{X}$. If $Q > 0$ wherever $P > 0$, the $f$-divergence between $P$ and $Q$ is defined as $D_f(P\|Q) = \sum_{\mathbf{x}} Q(\mathbf{x}) f\left(\frac{P(\mathbf{x})}{Q(\mathbf{x})}\right)$.

Commonly used $f$-divergences include the Kullback-Leibler divergence, $\chi^2$-divergence, and total variation distance. The total variation distance is especially relevant to our results.

**Definition 2.5** (Total variation distance). The *total variation distance (TVD)* between two probability distributions $P$ and $Q$ over a set of $n$ Boolean variables $\mathbf{X}$ is defined as $D_{\mathsf{TV}}(P\|Q) = \frac{1}{2} \sum_{\mathbf{x} \in \mathbf{X}} |P(\mathbf{x}) - Q(\mathbf{x})|$, or equivalently $D_{\mathsf{TV}}(P\|Q) = \max_{S \subseteq \{0,1\}^n} |P(S) - Q(S)|$.

We introduce the following notion to describe probabilistic models that approximate distributions within some bounded distance.

**Definition 2.6** ($\epsilon$-$D$-Approximation). Let $P, Q$ be two probability distributions and $D$ be a distance measure between distributions. We say that $Q$ is an $\epsilon$-$D$-*approximator* of $P$ if $D(P\|Q) < \epsilon$ for some $\epsilon > 0$.

For instance, we refer to a probabilistic circuit $Q$ that approximates our target distribution $P$ such that $D_{\mathsf{TV}}(P\|Q) < \epsilon$ a $\epsilon$-$\mathcal{D}_{\mathsf{TV}}$-*approximator*. The majority of our results are derived using properties of the total variation distance, due to its nice properties as a distance metric. To extend our results to other $f$-divergences, we utilize the following class, which provides an upper bound on the total variation distance.

**Definition 2.7** ($k$-convex $f$-divergence [34]). A $\mathbb{R} \cup \{\infty\}$-valued function $f$ on a convex set $K \subseteq \mathbb{R}$ is $k$-convex if $x, y \in K$ and $t \in [0, 1]$ implies $f((1-t)x + ty) \leq (1-t)f(x) + tf(y) - kt(1-t)\frac{(x-y)^2}{2}$. When $f$ is twice differentiable, this is equivalent to $f''(x) \geq k$ for all $x \in K$. In the case that $k = 0$ this reduces to the normal notion of convexity. An $f$-divergence $D_f$ is $k$-*convex* over an interval $K$ for $k \geq 0$ if the function $f$ is $k$-convex on $K$.

We provide a table in Appendix A.1 summarizing which $f$-divergence measures are $k$-convex and for which value of $k$. Throughout this paper, we express approximation bounds using $k$-convex $f$-divergences as they naturally encapsulate bounds on many common distance measures. For instance, for any $k$-convex $f$-divergence between $P$ and $Q$, we have that $D_{\mathsf{TV}}(P\|Q)^2 < D_f(P\|Q)/k$ [34]. As we will see later, the bounds on the TV distance can naturally be connected to guarantees for approximation inference. For KL-divergence, which is the most commonly used objective for learning probabilistic models, we can use Pinsker's inequality [45] to obtain $D_{\mathsf{TV}}(P\|Q) < \sqrt{\frac{1}{2} D_{\mathsf{KL}}(P\|Q)}$.

## 3 Approximate Modeling with Tractable Marginals is NP-hard

Most works characterizing the expressive efficiency of different circuit classes have been concerned with *exact* representations [19, 2, 8, 48, 49, 20]. While Chubarian and Turán [12] and De Colnet and Mengel [21] have recently studied the ability (and hardness) of logical circuit classes to compactly *approximate* Boolean functions, to the best of our knowledge, our results are the first to show hardness of *compactly approximating probability distributions* using different families of tractable PCs.

As discussed previously, the complexity of approximately modeling distributions with PCs is valuable for understanding: (1) potential limitations in the hypothesis space of PC learning algorithms, and (2) the feasibility of approximate inference with guarantees through approximate compilation. This section aims to answer this focusing on probabilistic models that are tractable for marginal queries. We first show that a form of approximate marginal inference using this scheme requires a non-trivial bound on the total variation distance between the target distribution and the approximate model, and next prove that finding such an approximator is NP-hard.

### 3.1 Relative Approximation of Marginals

We consider *relative approximation*[4] of marginal queries. Let $P(\mathbf{X})$ be a probability distribution over a set of variables $\mathbf{X}$. Then we say another distribution $Q(\mathbf{X})$ is a *relative approximator* of

---

[4]Also called multiplicative approximation or approximation within a factor.

marginals of $P$ w.r.t. $0 \le \epsilon \le 1$ if: $\frac{1}{1+\epsilon} \le \frac{P(\mathbf{y})}{Q(\mathbf{y})} \le 1 + \epsilon$ for every assignment $\mathbf{y}$ to subset of variables $Y \subseteq \mathbf{X}$. Relative approximation is often considered for approximate inference of graphical models and the closely related approximate (weighted) model counting [23, 3]. We first show that relative approximation for all marginal queries implies a non-trivially bounded total variation distance.

**Theorem 3.1** (Relative approximation implies bounded $D_{\mathsf{TV}}(P\|Q)$). *Let $\epsilon > 0$ and $P, Q$ be two probability distributions over $\mathbf{X}$. If $Q$ is a relative approximator of marginals for $P$, then $D_{\mathsf{TV}}(P\|Q) \le \frac{\epsilon}{2}$.*

*Proof.* As $Q$ is a relative approximator of $P$, for all assignment $\mathbf{x}$ we have that $\frac{1}{1+\epsilon} \le \frac{P(\mathbf{x})}{Q(\mathbf{x})} \le 1 + \epsilon$ which implies $|P(\mathbf{x}) - Q(\mathbf{x})| \le \epsilon \min(P(\mathbf{x}), Q(\mathbf{x}))$. Therefore, $D_{\mathsf{TV}}(P\|Q) = \frac{1}{2} \sum_{\mathbf{x}} |P(\mathbf{x}) - Q(\mathbf{x})| \le \frac{1}{2} \sum_{\mathbf{x}} \epsilon \min(P(\mathbf{x}), Q(\mathbf{x})) \le \frac{\epsilon}{2}$. $\square$

In other words, $D_{\mathsf{TV}}(P\|Q) \le \epsilon/2$ is a necessary condition for $Q$ to be a relative approximator of marginals of $P$ w.r.t. $\epsilon$. However, it is still not a sufficient condition as shown below.

**Proposition 3.2** (Bounded $D_{\mathsf{TV}}(P\|Q)$ does not imply relative approximation). *There exists a family of distributions $P$ that have $\epsilon$-$D_{\mathsf{TV}}$-approximators, yet for any such approximator $Q$, the relative approximation error of marginals between $P$ and $Q$ can be arbitrarily large.*

We prove the above proposition by explicitly constructing a family of distributions $\mathcal{Q}$ such that every $Q \in \mathcal{Q}$ is an $\epsilon$-$D_f$-approximator for any arbitrary $\epsilon > 0$ and distribution $P$ yet $P(\mathbf{x})/Q(\mathbf{x})$ can be arbitrarily large for some $\mathbf{x}$. See Appendix A.3.1 for the full construction.

It is known that relative approximation of marginals is NP-hard for Bayesian networks [14].[5] Thus, it immediately follows that approximately representing arbitrary distributions using polynomial-sized PCs tractable for marginals (e.g., decomposable PCs) such that the PC is a relative approximator of all marginals is also NP-hard. However, approximating *all* marginal queries is quite a strong condition, and we may still want to closely approximate distributions as they could be useful in approximating *some* marginal queries. In particular, because approximating a distribution $P$ with a bounded TV distance is a necessary but not sufficient condition for the NP-hard problem of relative approximation of marginals, this raises the question whether it is still possible to efficiently approximate the distribution $P$ with a compact PC $Q$ that is tractable for marginals. Unfortunately, we next answer this in the negative.

## 3.2 Hardness of Approximating Distributions using Tractable Models for Marginals

We consider approximating potentially *unnormalized* distributions, which can be considered a generalization of probability distributions by omission of the normalizing constant.

**Definition 3.3** (Unnormalized Distributions). Any (unnormalized) distribution $\hat{P} : \mathrm{val}(\mathbf{X}) \to \mathbb{R}$ must satisfy the following: (1) $\hat{P}(\mathbf{x}) \ge 0$ for any $\mathbf{x}$; (2) The normalization constant $Z = \sum_{\mathbf{x} \in \mathbf{X}} \hat{P}(\mathbf{x})$ is well-defined and finite.

See from this definition that an unnormalized distribution can easily be converted to a probability distribution $P$ if $Z$ is computable in polynomial time: $P(\mathbf{x}) = \hat{P}(\mathbf{x})/Z$. Unnormalized distributions are relevant to the goal of tractable approximation, as many probabilistic models that we may want to approximate—including factor graphs [32] and energy-based models [44]—represent unnormalized distributions.

**Theorem 3.4** (Hardness of $D_f$-approximation). *Given a (potentially unnormalized) probability distribution $\hat{P}$ and a $k$-convex $f$-divergence $D_f$, for any $0 < \epsilon < \frac{1}{4}$, it is NP-hard to represent the $k\epsilon^2$-$D_f$-approximation of its normalized distribution $P$ as a model that can tractably compute marginals.*

*Proof.* We will prove the above using a reduction from SAT. Let $\hat{P}$ be a Boolean formula over $\mathbf{X} = \{X_1, \ldots, X_n\}$ and $\epsilon < \frac{1}{4}$. See that $\hat{P}$ satisfies all requirements to be considered an unnormalized

---

[5]In fact, there also exists no randomized polynomial time algorithm for relative approximation of marginals unless $\mathsf{RP} = \mathsf{NP}$ [14].

probability distribution. We then define a new Boolean formula $\hat{P}'$ over $\mathbf{X}$ and an auxiliary variable $Y$: $\hat{P}' = (Y \wedge \hat{P}) \vee (\neg Y \wedge X_1 \wedge \cdots \wedge X_n)$. Clearly $\hat{P}'$ has $\mathrm{MC}(\hat{P}) + 1$ models. Let us now define a uniform distribution $P$ over these models of $\hat{P}'$ (i.e., by normalizing $\hat{P}'$). Suppose that we can efficiently obtain a probability distribution $Q$ such that $D_f(P\|Q) < k\epsilon^2$, which in turn implies that $D_{\mathsf{TV}}(P\|Q) < \epsilon$. From the definition of total variation distance, $|P(Y=1) - Q(Y=1)| < \epsilon < \frac{1}{4}$.

By construction, if $\hat{P}$ is unsatisfiable, there is no satisfying model of $\hat{P}'$ such that $Y = 1$, and thus $P(Y = 1) = 0$ and $Q(Y = 1) < \frac{1}{4}$. Otherwise, if $\hat{P}$ is satisfiable, then there are $\mathrm{MC}(\hat{P})$ many satisfying model of $\hat{P}'$ setting $Y = 1$, and thus we have $\mathrm{MC}(\hat{P}) \geq 1$ and $P(Y = 1) = \frac{\mathrm{MC}(\hat{P})}{1+\mathrm{MC}(\hat{P})}$, implying

$$ Q(Y=1) > \frac{\mathrm{MC}(\hat{P})}{1 + \mathrm{MC}(\hat{P})} - \frac{1}{4} \geq \frac{1}{2} - \frac{1}{4} \geq \frac{1}{4}. $$

Therefore, $\hat{P}$ is satisfiable if and only if $Q(Y = 1) \geq \frac{1}{4}$. In other words, we can decide SAT if we can efficiently compute an $k\epsilon^2$-$D_f$-approximation as a model that supports tractable marginals. $\square$

The following corollary immediately follows from the above proof.

**Corollary 3.5.** *Given a (potentially unnormalized) probability distribution $\hat{P}$, for $0 < \epsilon < \frac{1}{4}$, it is* NP-*hard to represent the $\epsilon$-$D_{\mathsf{TV}}$-approximation of its normalized distribution $P$ as a model that can tractably compute marginals.*

Thus, the class of polynomial-sized probabilistic models supporting tractable marginals cannot contain $k\epsilon^2$-$\mathcal{D}_f$-approximations for all distributions unless P = NP. This includes decomposable PCs [8], sum of squares circuits [31], probabilistic generating circuits [50], determinantal point processes [1], Inception PCs [47], and positive unital circuits [22]. Furthermore, by Pinsker's inequality this suggests that there exists distributions for which obtaining a decomposable PC with bounded KL-divergence of $1/8$ is NP-hard. While Martens and Medabalimi [33] previously showed a related result that there exists a function for which a sequence of decomposable PCs converging to approximate the function arbitrarily well requires an exponential size, our result applies more broadly to any class of models supporting tractable marginals as well as allowing for a bounded *but non-vanishing* approximation error. Thus, it is difficult not only to exactly represent functions or distributions as compact decomposable PCs, but also to approximate within some small distance. Consequently, this presents a major challenge to the idea of using approximate compilation of PCs for approximate inference if the desired error tolerance $\epsilon$ is sufficiently small.

## 4 Large Deterministic & Decomposable PCs for Approximate Modeling

Continuing our characterization of the approximation power of PCs, we now turn to the family of deterministic and decomposable PCs. We take inspiration from related results for logical circuits. In particular, Bova et al. [2] proved an exponential separation between DNNFs and d-DNNFs: i.e., there is a family of Boolean functions that can be compactly represented by decomposable circuits but requires exponentially sized deterministic and decomposable circuits. Furthermore, De Colnet and Mengel [21] showed that there exist functions that require exponential size to approximate with d-DNNFs under two notions of approximation for Boolean functions.

Nevertheless, this does not immediately imply the same separation for probabilistic circuits due to two key reasons: (1) approximation for PCs is measured in terms of divergences between distributions rather than some probabilistic error between Boolean functions, and (2) our approximator can represent arbitrary distributions instead of being limited to a Boolean function (or a uniform distribution over it). This section presents our proof of exponential separation between the class of decomposable PCs and that of deterministic and decomposable ones, by constructing a family of distributions that can be represented by compact decomposable PCs, but any PC that is also deterministic and approximates it within a bounded TV distance must have an exponential size.

We consider the *Sauerhoff function* [41] which was used to show the separation between DNNFs and d-DNNFs for exact compilation [2]. Let $g_n : \{0,1\}^n \to \{0,1\}$ be a function evaluating to 1 if and only if the sum of its inputs is divisible by 3. The *Sauerhoff function* is defined as $S_n :$

$\{0,1\}^{n^2} \to \{0,1\}$ over the $n \times n$ matrix $X = (x_{ij})_{1 \le i,j \le n}$ such that $S_n(X) = R_n(X) \vee C_n(X)$, where $R_n, C_n : \{0,1\}^{n^2} \to \{0,1\}$ are defined as $R_n(X) = \bigoplus_{i=1}^{n} g_n(x_{i,1}, x_{i,2}, \ldots, x_{i,n})$ and $C_n(X) = R_n(X^T)$. Here, $\oplus$ represents addition modulo 2.

There exists a DNNF of size $O(n^2)$ that exactly represents the Sauerhoff function $S_n$, constructed as a disjunction of two compact ordered binary decision diagrams (OBDDs)—a more restrictive kind of deterministic and decomposable circuits—that represent $R_n$ and $C_n$, respectively [2, Proposition 7]. We then define our family of target distributions $P_n$ as follows: let $\mathcal{C}_n$ be a DNNF for $S_n$ with size $O(n^2)$; then $P_n$ is a decomposable PC obtained by replacing the literals of $\mathcal{C}_n$ with corresponding indicator functions, $\wedge$ with $\otimes$, and $\vee$ with $\oplus$ nodes with uniform parameters, followed by smoothing the circuit.[6] Note that $P_n$ outputs a positive value on an input $\mathbf{x}$ if and only if $S_n(\mathbf{x}) = 1$. We will show that a deterministic and decomposable PC approximating $P_n$ requires exponential size.

**Theorem 4.1** (Exponential-size deterministic PC)**.** *A deterministic and decomposable PC that is a $\epsilon$-$D_{\text{TV}}$-approximator of $P_n$ for some $\epsilon \le \frac{1}{16} - \Omega(1/Poly(n^2))$ has size $2^{\Omega(n)}$.*

We will prove the above by first showing that approximation of $P_n$ with a deterministic and decomposable PC implies a form of *weak approximation* [21] of $S_n$ with a d-DNNF of the same size, and next proving that such d-DNNF must be exponentially large.

**Definition 4.2** (Weak approximation [21])**.** A Boolean formula $g$ is a weak $\epsilon$-approximation of another Boolean formula $f$ if $\text{MC}(f \wedge \neg g) + \text{MC}(\neg f \wedge g) \le \epsilon \cdot 2^n$.

**Proposition 4.3** (Bounded $D_{\text{TV}}$ implies weak approximation)**.** *Let $0 \le \epsilon < \frac{1}{8}$ and $P$ be a uniform distribution whose support is given by a Boolean function $f$. Suppose that $Q$ is a deterministic and decomposable PC representing an $\epsilon$-$D_{\text{TV}}$-approximator of $P$. Then there exists a d-DNNF $g$ which has size polynomial in the size of $Q$ that represents a $4\epsilon$-weak-approximator of $f$.*

*Proof.* Suppose that $P$ is a uniform distribution over the support given by a Boolean function $f$, and $Q$ a probability distribution such that $D_{\text{TV}}(P\|Q) < \epsilon < \frac{1}{8}$. Then, we can construct an (unnormalized) deterministic and decomposable PC $Q'$ by pruning the edges of $Q$ such that any assignment $\mathbf{x}$ is eliminated from the resulting support of $Q'$ if and only if $Q(\mathbf{x}) < \frac{1}{2^{n+1}}$. We call this support $g$. We provide this pruning algorithm, which relies on determinism as a key property, in Appendix A.3.2. Clearly, the size of $Q'$ is at most the size of $Q$. We will now briefly summarize how this pruning scheme induces a weak approximation and refer to Appendix A.3.3 for the full derivation. Intuitively, we know that the support of $Q$ must cover most of the support of $P$, as $P$ is a uniform distribution and $D_{\text{TV}}(P\|Q)$ is bounded. While it is possible for the support of $Q$ to be much larger than the support of $P$, then the probability assigned by $Q$ to assignments outside of the support of $P$ must also be very small to maintain a small TV distance. Thus, pruning away these assignments with small probability allows us to retrieve $Q'$ whose support has only a small number of non-overlapping assignments with models of $f$. More formally, we know that by our pruning scheme, $Q(\mathbf{x}) \ge \frac{1}{2^{n+1}}$ on $g$ and $Q(\mathbf{x}) < \frac{1}{2^{n+1}}$ on $\neg g$. Using this fact, we can lower bound our original total variation distance by $\frac{1}{2}(\text{MC}(f \wedge \neg g)/2^{n+1} + \text{MC}(\neg f \wedge g)/2^{n+1})$, which implies that $\text{MC}(f \wedge \neg g) + \text{MC}(\neg f \wedge g) < 4\epsilon \cdot 2^n$. Thus, $g$ is a $4\epsilon$-weak-approximator of $f$, and we can represent it as a polynomially sized d-DNNF by taking the deterministic and decomposable PC $Q'$ and converting it to a logical circuit. $\square$

**Proposition 4.4** (d-DNNF approximating $S_n$ has exponential size)**.** *A d-DNNF representing a $(\frac{1}{4} - \Omega(1/Poly(n^2)))$-weak-approximation of $S_n$ has size $2^{\Omega(n)}$.*

*Proof.* Let $\mathcal{C}$ be a d-DNNF such that it is a $(\frac{1}{4} - \Omega(1/Poly(n^2)))$-weak-approximation of $S_n$. Sauerhoff [41] showed that any "two-sided" rectangle approximation[7] (which matches the notion of weak approximation) of $S_n$ within $\frac{1}{4} - \Omega(1/Poly(n^2))$ must have size $2^{\Omega(n)}$. Bova et al. [2] further showed that a d-DNNF $\mathcal{C}$ computing a function $f$ is a balanced rectangle partition of $f$ with size at most $|\mathcal{C}|$. Thus, $\mathcal{C}$ must have size $2^{\Omega(n)}$. $\square$

We are now ready to prove our main result about exponential size lower bound on deterministic and decomposable PCs as approximators.

---

[6]Smoothing a decomposable PC takes polynomial (worst-case quadratic) time [8].

[7]See Appendix A.2 for details on rectangle partitions.

*Proof of Theorem 4.1.* Suppose that we have a deterministic and decomposable PC $Q$ that is an $\epsilon$-$D_{\text{TV}}$-approximator of $P_n$, where $\epsilon = (\frac{1}{16} - \Omega(1/Poly(n^2)))$. Consider then the uniform distribution $U$ over $S_n$. Then, by the triangle inequality, $D_{\text{TV}}(U\|Q) \leq D_{\text{TV}}(U\|P_n) + D_{\text{TV}}(P_n\|Q) < D_{\text{TV}}(U\|P_n) + \epsilon$. By Bova et al. [2, Proposition 7], we know that the DNNF constructed to represent the Sauerhoff function is a disjunction of two OBDDs, which respectively represent $R_n, C_n$. As each OBDD can easily be translated to a d-NNF with a polynomial size increase, their PC counterparts (OR to $\oplus$ and AND to $\otimes$) will still represent the same Boolean functions [6]. Thus, the non-deterministic PC representing $P_n$ based on this construction only has one non-deterministic sum node at the root, and can return values at most 2. This allows us to see that $D_{\text{TV}}(U\|P) \leq \left| \frac{1}{|S_n|} - \frac{2}{|S_n|+1} \right| < \frac{1}{|S_n|} < \eta$ where $\eta = \frac{1}{(1-1/\sqrt{2})2^{n^2}}$ based on the fact that $|S_n| > (1-\beta)2^{n^2}$ for $\beta < 1/\sqrt{2}$, derived from the low 0-density property of $S_n$ under the uniform distribution [41]. Therefore, $D_{\text{TV}}(U\|Q) < \epsilon + \eta$. Since, the $\Omega(1/Poly(n^2))$ term in $\epsilon$ subsumes $\eta$, we can more simply say $D_{\text{TV}}(U\|Q) < \frac{1}{16} - \Omega(1/Poly(n^2))$. Next, we construct a d-DNNF $\mathcal{C}'$ from $Q$ by replacing indicators with literals, $\oplus$ with $\vee$, and $\otimes$ with $\wedge$. By Proposition 4.3, $\mathcal{C}'$ is a $(\frac{1}{4} - \Omega(1/Poly(n^2)))$-weak-approximation of $S_n$. Thus, by Proposition 4.4, $|Q| = |\mathcal{C}'| = 2^{\Omega(n)}$. $\qquad\square$

To sum up, we constructed a decomposable PC $P_n$ that has size $O(n^2)$ such that any deterministic and decomposable PC approximating it has size $O(2^{\Omega(n)})$, thereby showing an unconditional exponential gap for approximation between decomposable PCs and deterministic and decomposable PCs. This result highlights a fundamental limitation: approximate modeling does not grant us an additional flexibility to overcome exponential expressive efficiency gaps. Moreover, we next show that approximate modeling, even when somehow obtained, is unfortunately still not enough to use PCs for efficient approximate inference with guarantees in general.

## 5   Relationship between Approximate Modeling and Inference

We have shown that even if we allow some approximation error, it is hard to efficiently approximate distributions using tractable probabilistic circuits. Given that approximate modeling remains a hard task, one would hope that computing the approximators with bounded distance would allow us to approximate hard inference queries with bounded error. In this section, we study the relationship between approximate modeling and inference, in particular focusing on *relative* and *absolute* approximations of *marginal*, *conditional*, and *maximum-a-posteriori (MAP)* queries.

In Section 3.1, we showed that bounded total variation distance is a necessary but not sufficient condition for relative approximation of marginals. We now consider a slightly weaker notion of approximation called *absolute approximation*. Let $P(\mathbf{X})$ be a probability distribution over a set of variables $\mathbf{X}$. Then we say another distribution $Q(\mathbf{X})$ is an *absolute approximator* of marginals of $P$ with respect to $0 \leq \epsilon \leq 1$ if: $|P(\mathbf{y}) - Q(\mathbf{y})| \leq \epsilon$ for every assignment $\mathbf{y}$ to a subset of variables $Y \subseteq \mathbf{X}$. We show that any model that is a $k\epsilon^2$-$D_f$-approximator of $P$ must also be an absolute approximator of marginals of $P$ with respect to $\epsilon$.

**Theorem 5.1** (Bounded $D_f$ implies absolute approximation of marginals). *Given two distributions $P$ and $Q$ over a set of variables $\mathbf{X}$ and $0 \leq \epsilon \leq 1$, if $D_f(P\|Q) < k\epsilon^2$ then for all assignments $\mathbf{y}$ to a subset $\mathbf{Y} \subseteq \mathbf{X}$, we have $|P(\mathbf{y}) - Q(\mathbf{y})| < \epsilon$.*

*Proof.* Note that while the absolute error of marginals is symmetric between $P$ and $Q$, $f$-divergence between $P$ and $Q$, such as the KL-divergence, is not symmetric. Therefore, we utilize the implications derived in [34], that $D_f(P\|Q) < k\epsilon^2$ then $D_{\text{TV}}(P\|Q) < \epsilon$. Moreover, given that the total variation distance is an $f$-divergence, we know that by the monotonicity property [36] $D_f(P(\mathbf{Y}), Q(\mathbf{Y})) \leq D_f(P(\mathbf{X}), Q(\mathbf{X}))$ for any $\mathbf{Y} \subseteq \mathbf{X}$. By definition, $\max_{S \subseteq \{0,1\}^n} |P(S) - Q(S)| < \epsilon$, and thus $\forall \mathbf{y} : |P(\mathbf{y}) - Q(\mathbf{y})| < \epsilon$. $\qquad\square$

From the above proof, we also immediately derive the following corollary.

**Corollary 5.2** (Bounded $D_{\text{TV}}$ implies absolute approximation of marginals). *Given two distributions $P$ and $Q$ over a set of variables $\mathbf{X}$ and $0 \leq \epsilon \leq 1$, if $D_{\text{TV}}(P\|Q) < \epsilon$, then for all assignments $\mathbf{y}$ to $\mathbf{Y} \subseteq \mathbf{X}$, we have $|P(\mathbf{y}) - Q(\mathbf{y})| < \epsilon$.*

Since marginals are tractable for decomposable PCs, approximating a target distribution with bounded $f$-divergence using a decomposable PC implies that marginals can be approximated in polynomial-time with bounded absolute error. This aligns with Dagum and Luby [14], who showed that there exists a randomized polynomial-time algorithm for the absolute approximation of marginals for Bayesian networks.

We next study approximate inference implications for MAP inference. Let $P(\mathbf{X})$ be a probability distribution over a set of variables $\mathbf{X}$. We say another distribution $Q(\mathbf{X})$ is an absolute approximator of the *maximum-a-posteriori* of $P$ with respect to $0 \leq \epsilon \leq 1$ if: for every assignment $\mathbf{e}$ (called the evidence) to a subset $\mathbf{E} \subseteq \mathbf{X}$, $|\max_{\mathbf{y}} P(\mathbf{y}, \mathbf{e}) - \max_{\mathbf{y}} Q(\mathbf{y}, \mathbf{e})| \leq \epsilon$ where $\mathbf{Y} = \mathbf{X} \setminus \mathbf{E}$. We next show that $k\epsilon^2$-$D_f$-approximators are also absolute approximators of MAP with respect to $\epsilon$.

**Theorem 5.3** (Bounded $D_f$ implies absolute approximation of MAP). *Given two distributions $P$ and $Q$ over a set of variables $\mathbf{X}$ and $0 \leq \epsilon \leq 1$, if $D_f(P\|Q) < k\epsilon^2$ then for every assignment $\mathbf{e}$ to a subset $\mathbf{E} \subseteq \mathbf{X}$, we have $|\max_{\mathbf{y} \in \mathbf{Y}} P(\mathbf{y}, \mathbf{e}) - \max_{\mathbf{y} \in \mathbf{Y}} Q(\mathbf{y}, \mathbf{e})| < \epsilon$ where $\mathbf{Y} = \mathbf{X} \setminus \mathbf{E}$.*

*Proof.* Analogous to Theorem 5.1, we utilize the fact that if we have $D_f(P\|Q) < k\epsilon^2$, we know $D_{\mathsf{TV}}(P\|Q) < \epsilon$. Using $D_{\mathsf{TV}}(P\|Q) < \epsilon$, we have $\max_{\mathbf{x}} |P(\mathbf{x}) - Q(\mathbf{x})| < \epsilon$ by definition. Then for all $\mathbf{x}$, $Q(\mathbf{x}) - \epsilon < P(\mathbf{x}) < Q(\mathbf{x}) + \epsilon$. W.l.o.g., suppose $\max P(\mathbf{x}) > \max Q(\mathbf{x})$. Then $|\max P(\mathbf{x}) - \max Q(\mathbf{x})| < (\max Q(\mathbf{x}) + \epsilon) - \max Q(\mathbf{x}) = \epsilon$. Thus, as this holds for all $\mathbf{x}$, we can extend this to $|\max_{\mathbf{y} \in \mathbf{Y}} P(\mathbf{y}, \mathbf{e}) - \max_{\mathbf{y} \in \mathbf{Y}} Q(\mathbf{y}, \mathbf{e})| < \epsilon$ for every assignment $\mathbf{e}$ to a subset $\mathbf{E} \subseteq \mathbf{X}$ and $\mathbf{Y} = \mathbf{X} \setminus \mathbf{E}$. Thus, approximating with bounded $f$-divergence by a deterministic and decomposable PC implies polynomial-time approximation of MAP with bounded absolute error. $\square$

Again, we can restrict this to the special case of total variation distance via the above.

**Corollary 5.4** (Bounded $D_{\mathsf{TV}}$ implies absolute approximation of MAP). *Given two distributions $P$ and $Q$ over a set of variables $\mathbf{X}$ and $0 \leq \epsilon \leq 1$, if $D_{\mathsf{TV}}(P\|Q) < \epsilon$ then for every assignment $\mathbf{e}$ to a subset $\mathbf{E} \subseteq \mathbf{X}$, we have $|\max_{\mathbf{y} \in \mathbf{Y}} P(\mathbf{y}, \mathbf{e}) - \max_{\mathbf{y} \in \mathbf{Y}} Q(\mathbf{y}, \mathbf{e})| < \epsilon$ where $\mathbf{Y} = \mathbf{X} \setminus \mathbf{E}$.*

Thus, a deterministic and decomposable PC that is an $\epsilon$-$D_{\mathsf{TV}}$-approximator of a distribution $P$ would imply that exact MAP inference w.r.t. this PC grants us tractable approximate MAP inference w.r.t. the original distribution $P$. However, the converse does not hold: a PC that can be used for approximate MAP inference is not necessarily a good approximation of the full distribution.

**Counterexample 1.** Consider a family of distributions $P(\mathbf{x})$ such that $\max_{\mathbf{x}} P(\mathbf{x}) < \epsilon$. Then, we construct a distribution $P'(\mathbf{X}, Z)$ such that $P(\mathbf{x}, Z = 1) = P(\mathbf{x})$ and $P(\mathbf{x}, Z = 0) = 0$. Similarly, let $Q(\mathbf{X}, Z)$ be such that $Q(\mathbf{x}, Z = 0) = P(\mathbf{x})$ and $Q(\mathbf{x}, Z = 1) = 0$. Thus, for any assignment $\mathbf{e}$ to $\mathbf{E} \subseteq \mathbf{X} \cup \{Z\}$, $|\max_{\mathbf{y}} P(\mathbf{y}, \mathbf{e}) - \max_{\mathbf{y}} Q(\mathbf{y}, \mathbf{e})| \leq P(\mathbf{x}) < \epsilon$, so $Q$ is an absolute approximator of the MAP of $P$. However, $D_{\mathsf{TV}}(P\|Q) = 1$ as $P$ and $Q$ have disjoint supports.

Lastly, not all tractable queries for PCs are guaranteed to admit absolute approximation even under this framework of approximate modeling with bounded distance.

**Theorem 5.5** (Bounded $D_{\mathsf{TV}}$ does not imply absolute approx. of conditionals/conditional MAP). *There exists a family of distributions $P$ that have $\epsilon$-$D_{\mathsf{TV}}$- approximators, yet the absolute approximation for conditional marginals and conditional MAP can be arbitrarily large.*

*Proof.* Let $P$ be a probability distribution over $\mathbf{X}$ such that $P(\mathbf{e}) < 1/k$ for some assignment $\mathbf{e}$ to $\mathbf{E} \subseteq \mathbf{X}$. Let $\mathbf{Y} = \mathbf{X} \setminus \mathbf{E}$. We construct another distribution $Q$ such that: $Q(\mathbf{y}^*, \mathbf{e}) = P(\mathbf{e})P(\mathbf{y}^*|\mathbf{e}) + k\epsilon P(\mathbf{e})$ where $\mathbf{y}^*$ maximizes $P(\mathbf{y}|\mathbf{e})$; $Q(\mathbf{y}_1, \mathbf{e}) = P(\mathbf{e})P(\mathbf{y}_1|\mathbf{e}) - k\epsilon P(\mathbf{e})$ for another assignment $\mathbf{y}_1$; and $Q(\mathbf{x}) = P(\mathbf{x})$ for all other assignments $\mathbf{x}$. Note that $Q$ is normalized by construction. Then the total variation distance between $P$ and $Q$ is: $D_{\mathsf{TV}}(P\|Q) = \frac{1}{2}(|P(\mathbf{y}^*, \mathbf{e}) - Q(\mathbf{y}^*, \mathbf{e})| + |P(\mathbf{y}_1, \mathbf{e}) - Q(\mathbf{y}_1, \mathbf{e})|) = k\epsilon P(\mathbf{e}) < \epsilon$. On the other hand, the absolute approximation error of conditional MAP can grow arbitrarily by increasing $k$: $|\max_{\mathbf{y}} P(\mathbf{y}|\mathbf{e}) - \max_{\mathbf{y}} Q(\mathbf{y}|\mathbf{e})| = |P(\mathbf{y}^*|\mathbf{e}) - P(\mathbf{y}^*|\mathbf{e}) - k\epsilon| = k\epsilon$. Note that the same also holds for absolute approximation of conditionals. $\square$

Moreover, because relative approximation of conditionals imply their absolute approximation [14], bounded $D_{\mathsf{TV}}$ also does not imply relative approximation of conditionals. It is well known that absolute and relative approximation of conditionals is NP-hard in Bayesian networks [14]. Even though approximate modeling with a bounded $D_{\mathsf{TV}}$ is also an NP-hard task, solving it still does not

guarantee a polynomial-time algorithm for approximating conditional queries. This highlights a key limitation: while tractability of queries is guaranteed by the structural properties of our learned PCs, some queries do not yield "good" approximations for all assignments even after learning within bounded distance.

# 6    Conclusion and Discussions

We established the hardness of approximating distributions with tractable probabilistic models such that the $f$-divergence is small. First, we showed that this task is NP-hard for any model supporting tractable marginal inference, including decomposable PCs. Then, we used the Sauerhoff function to demonstrate an exponential size gap between the class of decomposable PCs and that of deterministic and decomposable PCs when allowing for a bounded approximation error. This proves that the expressive efficiency gap that exists in exact compilation persists even under the relaxed approximation conditions. Finally, we characterized which queries remain well-approximated under the framework of approximate compilation.

These results highlight key challenges in learning compact and expressive PCs while maintaining tractable inference. In light of this, we ask: can a polynomial-time algorithm enable learning an $\epsilon$-approximator for a broad family of distributions with a more relaxed $\epsilon$? While this is trivial when total variation is near 1, investigating whether structurally constrained PCs remain expressive under weaker approximation could reveal key limits of learnability. Furthermore, does there exist modeling conditions that are sufficient to guarantee relative approximation of various queries? Lastly, we see this work as a first step to encourage further theoretical studies on approximate modeling and inference with guarantees using tractable models. In this paper, we focused on PCs that are tractable for marginal and MAP queries, but there are large classes of tractable models whose efficient approximation power remain largely unknown.

## Acknowledgments and Disclosure of Funding

We thank the anonymous reviewers for their feedback towards improving this paper. We also thank Adrian Ciotinga for helpful feedback and discussion. This work was supported in part by the AFOSR grant FA9550-25-1-0320.

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

# A Appendix

## A.1 $k$-convex $f$-divergences

Table 1: Examples of $k$-convex $f$-divergences [34]

| Divergence | $f$ | $k$ | Domain |
|---|---|---|---|
| Kullback-Leibler | $t \log t$ | $\frac{1}{M}$ | $(0, M]$ |
| Total variation | $\frac{|t-1|}{2}$ | $0$ | $(0, \infty)$ |
| Pearson's $\chi^2$ | $(t-1)^2$ | $2$ | $(0, \infty)$ |
| Squared Hellinger | $2(1 - \sqrt{t})$ | $M^{-\frac{3}{2}}/2$ | $(0, M]$ |
| Reverse KL | $-\log t$ | $\frac{1}{M^2}$ | $(0, M]$ |
| Vincze-Le Cam | $\frac{(t-1)^2}{t+1}$ | $\frac{8}{(M+1)^3}$ | $(0, M]$ |
| Jensen–Shannon | $(t+1)\log\frac{2}{t+1} + t \log t$ | $\frac{1}{M(M+1)}$ | $(0, M]$ |
| Neyman's $\chi^2$ | $\frac{1}{t} - 1$ | $\frac{2}{M^3}$ | $(0, M]$ |
| Sason's $s$ | $\log(s+t)^{(s+t)^2} - \log(s+1)^{(s+1)^2}$ | $2\log(s+M) + 3$ | $[M, \infty), s > e^{-3/2}$ |
| $\alpha$-divergence | $\frac{4\left(1 - t^{\frac{1+\alpha}{2}}\right)}{1 - \alpha^2}, \quad \alpha \neq \pm 1$ | $M^{\frac{\alpha-3}{2}}$ | $\begin{cases} [M, \infty), & \alpha > 3 \\ (0, M], & \alpha < 3 \end{cases}$ |

## A.2 Rectangle Partitions

Rectangle partitions are a powerful tool used in communication complexity to analyze the size of communication protocols. The main idea is to represent the communication protocol for a function $f : \{0,1\}^n \to \{0,1\}$ into a $2^n \times 2^n$ matrix $M_f$ where $M_f[\mathbf{x}, \mathbf{y}] = f(\mathbf{x}, \mathbf{y})$, then partition $M_f$ into a set of monochromatic rectangles which cover the input space of all possible pairs. Here, monochromatic means that a given rectangle covers only the outputs equal to 0 or 1, but not both. This allows us to derive lower bounds on the communication complexity of a function $f$. Furthermore, Bova et al. [2] showed the relation between rectangle covers and partitions to the size of DNNF and d-DNNF formulas. For all definitions below, assume that $\mathbf{X}$ is a finite set of variables.

We begin by describing *partitions* of $\mathbf{X}$, corresponding to our partition of $M_f$.

**Definition A.1** (Partition [2]). A partition of $\mathbf{X}$ is a sequence of pairwise disjoint subsets $\mathbf{X}_i$ of $\mathbf{X}$ such that $\bigcup_i \mathbf{X}_i = \mathbf{X}$. A partition of two blocks $(\mathbf{X}_1, \mathbf{X}_2)$ is *balanced* if $|\mathbf{X}|/3 \leq \min(|\mathbf{X}_1|, |\mathbf{X}_2|)$.

We can now define *rectangles*:

**Definition A.2** (Combinatorial Rectangle [2]). A rectangle over $\mathbf{X}$ is a function $r : \{0,1\}^{|\mathbf{X}|} \to \{0,1\}$ such that there exists and underlying partition of $\mathbf{X}$, called $(\mathbf{X}_1, \mathbf{X}_2)$ and functions $r_i : \{0,1\}^{|\mathbf{X}|} \to \{0,1\}$ for $i = 1, 2$ such that $r^{-1}(1) = r_1^{-1}(1) \times r_2^{-1}(1)$. A rectangle is *balanced* if the underlying partition is balanced.

Combining many of these rectangles together allows us to cover $M_f$, effectively covering the function $f$. The size of these covers provides lower bounds on the communication complexity of $f$.

**Definition A.3** (Rectangular Cover [2]). Let $f : \{0,1\}^{|\mathbf{X}|} \to \{0,1\}$ be a function. A finite set of rectangles $\{r_i\}$ over $\mathbf{X}$ is called a *rectangle cover* if

$$f^{-1}(1) = \bigcup_i r_i^{-1}(1).$$

The rectangle cover is referred to as a *rectangle partition* if the above union is disjoint. A rectangle cover is *balanced* if each rectangle in the cover is balanced.

To understand how these rectangle partitions relate to d-DNNFs and DNNFs, we utilize the following notions of *certificates* and *elimination*.

**Definition A.4** (Certificate [2]). Let $\mathcal{C}$ be a DNNF on $\mathbf{X}$. A *certificate of* $\mathcal{C}$ is a DNNF $T$ on $\mathbf{X}$ such that: $T$ contains the output gate of $\mathcal{C}$; if $T$ contains an $\wedge$-gate, $v$, of $\mathcal{C}$ then $T$ also contains every gate of $\mathcal{C}$ having an output wire to $v$; if $T$ contains an $\vee$-gate of $\mathcal{C}$, then $T$ also contains exactly one gate of $\mathcal{C}$ having an output wire to $v$. The output gate of $T$ coincides with the output gate of $\mathcal{C}$, and the gates of $T$ inherit their labels and wires from $\mathcal{C}$. We let $cert(\mathcal{C})$ denote the certificates of $\mathcal{C}$.

See from the above definition that

$$\mathcal{C}^{-1}(1) = \bigcup_{T \in cert(\mathcal{C})} T^{-1}(1).$$

This is useful tool in relation to rectangle partitions due to the fact that given a DNNF $\mathcal{C}$, $T \in cert(\mathcal{C})$ and gate $g$, then $\mathcal{C}_g^{-1}(1) = \bigcup_{T \in cert(\mathcal{C}_g)} T^{-1}(1)$ where $\mathcal{C}_g$ represents the sub-circuit $\mathcal{C}$ rooted at gate $g$. Then we can represent $\mathcal{C}_g^{-1}(1)$ as a rectangle which separates the variables in the sub-circuit $\mathcal{C}$ rooted at $g$. Using this in conjunction with the *elimination* operation gives us the ability to compute the size of our circuit using rectangles.

**Definition A.5** (Elimination [2]). Let $\mathcal{C}$ be a DNNF and $g$ be a non-input gate. Then,

$$\mathcal{C}_{\neg g}^{-1}(1) = \bigcup_{T \in cert(\mathcal{C}) \setminus cert(\mathcal{C}_g)} T^{-1}(1)$$

In the case of a d-DNNF, by determinism we can write $\mathcal{C}\neg g^{-1}(1) = \mathcal{C}^{-1}(1) \setminus \mathcal{C}_g^{-1}(1)$.

Next we provide a short description on the relationship between the size of rectangle covers and Boolean circuits; for the full detailed proofs see [2]. Effectively, start with a d-DNNF $\mathcal{C}$ over variables $\mathbf{X}$ which computes a function $f$. Then, construct $\mathcal{C}^{i+1} = \mathcal{C}^i_{\neg g_i}$ by eliminating $g_i \in \mathcal{C}^i$ until we hit $l \leq |\mathcal{C}|$ such that $\mathcal{C}^l \equiv 0$. It can be shown that $R_i = \mathcal{C}^i_{g_i}{}^{-1}(1)$ is a balanced rectangle over $\mathbf{X}$. The set $\{R_i | i = 0, \ldots, l-1\}$ is then a balanced rectangle partition of $\mathcal{C}$ since $\mathcal{C}^{i+1}_{\neg g_i}{}^{-1}(1) = \emptyset$. Therefore, we can represent the size of d-DNNFs representing functions as the size of a balanced rectangle partition over said function. This implies that an exponential size rectangle partition implies exponentially large d-DNNF.

## A.3 Complete Proofs

### A.3.1 Distribution Construction for Proposition 3.2

*Proof.* Let $A$ be an event such that $P(A) = \delta$ and for some $K > 0$,

$$Q(\mathbf{x}) = \frac{P(\mathbf{x})}{K}, \ \forall \mathbf{x} \in A.$$

Then, let $Q(\mathbf{x}) = \lambda P(\mathbf{x}), \ \forall \mathbf{x} \in A^c$ for some constant $\lambda$. To ensure that $Q$ is normalized, see that we must have $\sum_{\mathbf{x}} Q(\mathbf{x}) = \sum_{\mathbf{x} \in A} Q(\mathbf{x}) + \sum_{\mathbf{x} \in A^c} Q(\mathbf{x}) = 1$. Therefore, we must have:

$$1 = \sum_{\mathbf{x} \in A} Q(\mathbf{x}) + \sum_{\mathbf{x} \in A^c} Q(\mathbf{x}) = \sum_{\mathbf{x} \in A} \frac{P(\mathbf{x})}{K} + \sum_{\mathbf{x} \in A^c} \lambda P(\mathbf{x}) = \frac{\delta}{K} + \lambda(1 - \sum_{\mathbf{x} \in A} P(\mathbf{x}))$$

$$= \frac{\delta}{K} + \lambda(1 - \delta)$$

Hence $\lambda = \frac{1 - \delta/K}{1 - \delta}$ and thus we can define

$$Q(\mathbf{x}) = \begin{cases} \frac{P(\mathbf{x})}{K}, & \mathbf{x} \in A \\ \frac{1-\delta/K}{1-\delta} P(\mathbf{x}), & \mathbf{x} \in A^c \end{cases}$$

Therefore, we just need to check then that the $f$-divergence between $P$ and $Q$ must be bounded.

$$\sum_{\mathbf{x}} Q(\mathbf{x}) f\left(\frac{P(\mathbf{x})}{Q(\mathbf{x})}\right) = \sum_{A} Q(\mathbf{x}) f\left(\frac{P(\mathbf{x})}{Q(\mathbf{x})}\right) + \sum_{A^c} Q(\mathbf{x}) f\left(\frac{P(\mathbf{x})}{Q(\mathbf{x})}\right)$$

$$= \sum_{A} \frac{P(\mathbf{x})}{K} f\left(\frac{P(\mathbf{x})}{P(\mathbf{x})/K}\right) + \sum_{A^c} \frac{1 - \delta/K}{1 - \delta} P(\mathbf{x}) f\left(\frac{1 - \delta}{1 - \delta/K}\right)$$

$$= \frac{f(K)}{K} \sum_{A} P(\mathbf{x}) + \frac{1 - \delta/K}{1 - \delta} f\left(\frac{1 - \delta}{1 - \delta/K}\right) \sum_{A^c} P(\mathbf{x})$$

$$= \frac{\delta f(K)}{K} + \frac{1 - \delta/K}{1 - \delta} f\left(\frac{1 - \delta}{1 - \delta/K}\right) (1 - \delta)$$

$$= \frac{\delta f(K)}{K} + \left(1 - \frac{\delta}{K}\right) f\left(\frac{1 - \delta}{1 - \delta/K}\right)$$

See that as $\delta \to 0$, the above approaches $0 + f(1)$.

By definition of $f$-divergence, $f(1) = 0$. Thus, the $f$-divergence—including the total variation distance—between $P$ and $Q$ can be very small, approaching 0, while the relative approximation error stays at a constant factor $K$. □

### A.3.2 Pruning Deterministic PCs for the Proof of Proposition 4.3

Suppose that $Q$ is a deterministic, decomposable and smooth probabilistic circuit. Given $Q$, wish to prune its edges such that in the resulting (unnormalized) PC $Q'$, $\mathbf{x}$ is in the support of $Q'$ if and only if $Q(\mathbf{x}) < \frac{1}{2^{n+1}}$. We describe our pruning algorithm below.

First, we collect the an upper bound on each edge $(n, c)$ that is the largest probability obtainable by any assignment $\mathbf{x}$ that uses that edge (propagates non-zero value through the edge in the forward pass for $Q(\mathbf{x})$). We denote this $EB(n, c)$, which stands for the Edge-Bound. This can be done in linear time in the size of the circuit using the Edge-Bounds algorithm [10]. This allows us to safely prune any edge whose Edge-Bound falls below a given threshold; i.e., prune edge $(n, c)$ if $EB(n, c) < \frac{1}{2^{n+1}}$.

Note that pruning some edges may cause the edge bounds for remaining edges to be tightened. Thus, we will repeat this process until all $Q(\mathbf{x}) < \frac{1}{2^{n+1}}$ are pruned away. Upon completion of this process, we return back the new pruned circuit $Q'$.

We know that this algorithm halts as there can only be a finite number of $\mathbf{x}$ such that $Q(\mathbf{x}) < \frac{1}{2^{n+1}}$. Moreover, given that we have only deleted edges from a $Q$, our circuit $Q'$ is still a deterministic, decomposable and smooth probabilistic circuit and has size polynomial in the size of $Q$. We are also assured by determinism that if we prune a path $Q(\mathbf{x})$, there exists no other path that can evaluate $Q(\mathbf{x})$ [6]; thus all $Q(\mathbf{x}) < \frac{1}{2^{n+1}}$ are deleted. Furthermore, by the property that there is only one accepting path per assignment $\mathbf{x}$, we know that we do not unintentionally delete any $Q(\mathbf{x}) \geq \frac{1}{2^{n+1}}$.

### A.3.3 Connecting Total Variation Distance and Weak Approximation for the Proof of Proposition 4.3

Suppose that $P$ is a uniform distribution over the support given by a Boolean function $f$, and $Q$ a probability distributions over the support given by a Boolean function $h$. We look to analyze the total variation distance with respect to support $g$, which is taken from $Q'$.

$$D_{\mathsf{TV}}(P\|Q) = \frac{1}{2}\left(\sum_{x \models f \wedge h} |P(\mathbf{x}) - Q(\mathbf{x})| + \sum_{\mathbf{x} \models f \wedge \neg h} P(\mathbf{x}) + \sum_{\mathbf{x} \models \neg f \wedge h} Q(\mathbf{x})\right) < \epsilon$$

$$\implies \sum_{x \models f \wedge h} |P(\mathbf{x}) - Q(\mathbf{x})| + \sum_{\mathbf{x} \models f \wedge \neg h} P(\mathbf{x}) + \sum_{\mathbf{x} \models \neg f \wedge h} Q(\mathbf{x}) < 2\epsilon$$

We partition $h$ into the disjoint sets $g$ and $\neg g \wedge h$ (note that every model of $g$ is already a model of $h$).

$$\sum_{\mathbf{x} \models f \wedge g} \left|\frac{1}{\mathrm{MC}(f)} - Q(\mathbf{x})\right| + \sum_{\mathbf{x} \models f \wedge (\neg g \wedge h)} \left|\frac{1}{\mathrm{MC}(f)} - Q(\mathbf{x})\right| + \frac{\mathrm{MC}(f \wedge \neg h)}{\mathrm{MC}(f)} + \sum_{\mathbf{x} \models \neg f \wedge h} Q(\mathbf{x}) < 2\epsilon$$

The LHS of above inequality is again lower bounded by:

$$\sum_{\mathbf{x} \models f \wedge (\neg g \wedge h)} \left| \frac{1}{\text{MC}(f)} - Q(\mathbf{x}) \right| + \frac{\text{MC}(f \wedge \neg h)}{\text{MC}(f)} + \sum_{\mathbf{x} \models \neg f \wedge h} Q(\mathbf{x})$$

$$> \sum_{\mathbf{x} \models f \wedge (\neg g \wedge h)} \left| \frac{1}{2^n} - \frac{1}{2^{n+1}} \right| + \frac{\text{MC}(f \wedge \neg h)}{\text{MC}(f)} + \sum_{\mathbf{x} \models \neg f \wedge g} Q(\mathbf{x})$$

$$> \sum_{\mathbf{x} \models f \wedge (\neg g \wedge h)} \left| \frac{1}{2^n} - \frac{1}{2^{n+1}} \right| + \frac{\text{MC}(f \wedge \neg h)}{\text{MC}(f)} + \frac{\text{MC}(\neg f \wedge g)}{2^{n+1}}$$

as $\text{MC}(f) \leq 2^n$ and $Q(\mathbf{x}) < 1/2^{n+1}$ for every $\mathbf{x} \models \neg g \wedge h$. Thus,

$$2\epsilon > \sum_{\mathbf{x} \models f \wedge (\neg g \wedge h)} \left| \frac{1}{2^n} - \frac{1}{2^{n+1}} \right| + \frac{\text{MC}(f \wedge \neg h)}{\text{MC}(f)} + \frac{\text{MC}(\neg f \wedge g)}{2^{n+1}}$$

$$> \frac{\text{MC}(f \wedge (\neg g \wedge h))}{2^{n+1}} + \frac{\text{MC}(f \wedge \neg h)}{\text{MC}(f)} + \frac{\text{MC}(\neg f \wedge g)}{2^{n+1}}$$

$$> \frac{\text{MC}(f \wedge (\neg g \wedge h))}{2^{n+1}} + \frac{\text{MC}(f \wedge \neg h)}{2^{n+1}} + \frac{\text{MC}(\neg f \wedge g)}{2^{n+1}} = \frac{\text{MC}(f \wedge \neg g) + \text{MC}(\neg f \wedge g)}{2^{n+1}},$$

implying $\text{MC}(f \wedge \neg g) + \text{MC}(\neg f \wedge g) < 2^{n+2}\epsilon = 2^n(4\epsilon)$. Note that the last equality above is due to $\neg h$ implying $\neg g$.

