# OpenReview forum: "On the Hardness of Approximating Distributions with Tractable Probabilistic Models"
_NeurIPS.cc/2025/Conference — NeurIPS 2025 spotlight_

### Official Review · Reviewer_om4K · 2025-06-04

**Clarity:** 2
**Significance:** 3
**Originality:** 3
**Rating:** 5
**Confidence:** 3

**Summary:**

Probabilistic circuits (PCs) are a general class of distributions that enjoy linear time inference for several important query classes, such as computing marginal probabilities or MAP states, by enforcing structural properties. This work investigates what probability distributions can be approximated with guarantees using PCs. First, k-convex f-divergences, and in particular the special case of total variational distance (TVD), are established as a relevant criterion of approximation. Next, the paper investigates how a bound on the divergence of the distribution gives bounds on the approximation on the queries. In particular, when the TVD is bounded, the absolute error of the marginals (Thm 5.2) and MAP state (Thm 5.3) is also bounded, but not the relative error on the marginals (Prop 3.2) or the absolute error of conditionals (Thm 5.5). In the other direction, it’s shown that bounds on the relative error on the marginals lead to a bound on the f-divergence (Thm 3.1). Finally, it is shown that under bounded TVD, there is an exponential separation between arbitrary and decomposable PCs (Thm 3.3) and between decomposable and deterministic PCs (Thm 4.1).

**Questions:**

- Research on approximating weighted model counts [0] (very similar to approximating marginals on PCs) has shown that multiplicative $\epsilon$ approximations do not give you much. Indeed, it is #P-hard just as exact WMC. However, when moving to $\epsilon$ to $\epsilon,\delta$ approximations WMC becomes NP instead of #P [1] (still hard, but a very substantial difference). While reading the paper, I was wondering if similar could be taken for PCs?
- (Lines 331-336) Dagum and Luby showed a FPRAS for marginals is possible, but you consider absolute approximation (i.e., FPAS) which is not possible. So I don’t get what this paragraph is supposed to say? Approximating with bounded f-divergence isn’t tractable, so I don’t see how this is an “*easy form of approximation*”?
- On a more general note, the paper mostly contains negative results. I don’t have any problem with this but it begs the question what hope there is for approximations if they are still hard to construct and even cannot give approximation guarantees for many queries on them. Do the authors see any practical applications in the future?

[0]: Chakraborty, Supratik, Kuldeep S. Meel, and Moshe Y. Vardi. "Approximate model counting." *Handbook of Satisfiability*.

[1]: Chakraborty, Supratik, Kuldeep S. Meel, and Moshe Y. Vardi. "On the hardness of probabilistic inference relaxations." *Proceedings of the AAAI Conference on Artificial Intelligence*.

**Ethical Concerns:**

["NO or VERY MINOR ethics concerns only"]

**Final Justification:**

As the paper seems technically sound and I personally found the results sufficiently interesting, I recommend acceptance.

Reviewer VR5d raised some valid concerns on implicit assumptions, but I feel these have been addressed adequately by the authors. Unlike Reviewer VR5d, I do feel that PCs are an important class of distributions, due their unique trade-off in tractability vs expressivity.

**Limitations:**

yes

**Paper Formatting Concerns:**

No concerns.

**Quality:**

4

**Strengths And Weaknesses:**

**Strengths.** The paper contains interesting theoretical results on approximating probability distributions and approximating inference, fundamental problems in machine learning. I was able to follow the paper fairly well, despite its very technical nature. To the best of my knowledge, the results are novel. Some results in the paper are not too surprising given existing results on approximate probabilistic inference. That being said, I still think that it’s useful to establish them, as this paper could form a basis for further study into approximate PCs.

**Weaknesses**. I feel the presentation could have been better on some points. I could follow fairly well as I’m familiar with the PC literature, but some concepts like conditional MAP might not be known to outsiders and are not introduced in the paper. The narrative and the overall picture was sometimes a bit lost among all the technical arguments. In my opinion, a graphical summary of the various results between the different approximations between circuit and inference types could make the paper more accessible. Furthermore, the paper is framed for PCs, but results outside of Sec. 4 are mostly not specific to PCs and could be of broader interest (this was not clear to me from the introduction). The paper also discusses little related work, even though e.g. there is work approximating weighted model counting that is very related to approximating marginals (c.f. questions).

---

> ### Author Rebuttal · Authors · 2025-07-31
>
> Thank you for the constructive feedback. We will incorporate your suggestions to improve presentation in the revision. The additional page available for camera-ready versions upon acceptance would allow us to provide more background on different queries, add a graphical summary, and discuss more related work on approximate weighted model counting. In particular, we agree that our results outside of Section 4 are more broadly applicable beyond PCs, and we will revise to make this clearer. This would strengthen the significance of our contributions to a broader community.
>
> Q1. Thank you for the interesting suggestion. We had not considered probabilistic approximations for this paper, but this relaxation is a promising future direction. It could also be worthwhile to consider in conjunction with the average case complexity that Reviewer Leaf has brought to our attention.
>
> Q2. We agree that ''easy form of approximation'' is not exactly correct. More precisely, there exists a randomized algorithm for the absolute approximation of marginals. We still would not have absolute approximation from this in our sense, but this is one of the few types of approximation we discuss throughout our paper which even has a randomized algorithm. We will adjust the paragraph accordingly.
>
> Q3. While this paper mostly contains negative results, we hope that this will inspire future works on approximations to provide approximate inference guarantees, through relaxations such as probabilistic approximations or average-case guarantees or by considering a special case of target distributions.

---

> > ### Comment · Reviewer_om4K · 2025-08-01
> >
> > Thank you for the clarifications, I maintain my original assessment of the paper.

---

### Official Review · Reviewer_sZag · 2025-06-13

**Clarity:** 3
**Significance:** 3
**Originality:** 3
**Rating:** 5
**Confidence:** 3

**Summary:**

Probabilistic circuits (PCs) are appealing models because they support polynomial-time inference queries like marginalization under relatively mild structural assumptions. Their issue, however, is their limited expressive efficiency in the sense that there are probability distributions that cannot be encoded by a PC of polynomial size in the number of variables. This then motivates the problem of learning a circuit that *approximately* encodes the probability distribution under some distance metric such as the total variation distance, which in turn is bounded for any for any bounded $k$-convex $f$-divergence. The authors prove that it is unfortunately NP-hard to obtain such approximations of the probability distributions. The authors also show a gap in the expressive efficiency between the decomposable PCs and decomposable and deterministic PCs in an approximate setting, and study the relationship between the precision of the inference queries and the distance of the PC to the target distribution.

**Questions:**

To my understanding, total variation distance between two distributions is difficult to compute in general (and presumably the same holds for many other $f$-divergences). I would thus be curious to hear how the authors see the practical value of their work considering the intractability of evaluating the divergences when the probability distributions are given implicitly, e.g., as PCs.

**Ethical Concerns:**

["NO or VERY MINOR ethics concerns only"]

**Final Justification:**

I maintain my earlier stance the work gives nice results on an important problem for tractable probabilistic modeling. While Reviewer VR5d had some concerns about Theorem 3.3, they appear to be mostly resolved now, so I see no issue for acceptance.

**Limitations:**

yes

**Quality:**

3

**Strengths And Weaknesses:**

I think the present theoretical work studies an important problem of tractable probabilistic modelling and obtains nice novel results there, as discussed in the Summary. I would like to note, though, that the proof of your Theorem 3.3 resembles the proof of Proposition 1 in Appendix A.5 of Pote and Meel [1], so I would consider acknowledging that in some manner. Nevertheless, I think the paper would be worth citing since they study a related problem of whether there exists an efficient algorithm to test the closeness of two PC distributions.

Overall, I think the paper is well-written and I see no immediate weaknesses. As a minor comment, please carefully check the capitalizations in the References, since e.g. "Bayesian" is consistently written in lowercase there.

[1] Yash Pote, Kuldeep S. Meel:
Testing Probabilistic Circuits. NeurIPS 2021: 22336-22347

---

> ### Author Rebuttal · Authors · 2025-07-31
>
> Thank you for the constructive feedback.
>
> While Pote and Meel are concerned with a different problem setup, we agree that the proof technique of thresholding using total variation distance and the area are related to our work. Thank you for the pointer, and we will cite the work in the revision. Furthermore, we will adjust the citations to ensure they have the correct capitalization, thank you for catching that mistake.
>
> To answer your question: The total variation distance is indeed hard to compute for many probabilistic models. While some f-divergences such as the KL-divergence can be computed exactly between two PCs, this requires structural constraints that are more restrictive than those we consider in this paper. Nevertheless, note that upper-bounding the divergences suffices to apply our results. Thus, we hope that this work can motivate approximation/learning schemes that provide bounds on divergences which can derive some approximate inference bounds. Moreover, while we showed that approximate modeling with bounded f-divergences is NP-hard, the hope is that with further assumptions we can provide more guarantees that hold in practical settings.

---

> > ### Comment · Reviewer_sZag · 2025-08-01
> >
> > Thank you for the response. I'll maintain my score.

---

### Official Review · Reviewer_Leaf · 2025-06-23

**Clarity:** 4
**Significance:** 3
**Originality:** 4
**Rating:** 5
**Confidence:** 4

**Summary:**

This submission studies hardness of approximating probability distributions with probabilistic circuits (PCs). Prior work mostly studies *exact* representation of probability distributions with PCs. Approximation is considered with respect to f-divergences, which generalize KL divergence and TV distance.

The main contributions are:
- Demonstrating NP hardness (reduction from SAT) of approximating an arbitrary distribution with bounded f-divergence, when the model class supports efficient computation of marginal queries.
- Demonstrating an (unconditional) exponential separation between approximations using "decomposable" PCs and the even more restricted class of "decomposable and deterministic" PCs.
- Showing that if approximate PCs are found for a given distribution, then absolute approximation guarantees can be given for marginal and MAP queries on the PC with respect to the real distribution.

**Questions:**

My questions surround the ability to prove stronger or broader hardness results.

1. Have the authors considered proving some notion of average-case hardness to support the existing worst-case NP hardness? E.g., can you point to possible techniques / plans that could lead to proving that even randomly chosen distributions within some class are hard to approximate using a PC, with high probability over the random choice of distribution?
2. In line with average case hardness, have the authors considered using well-studied cryptographic assumptions (e.g., factoring, learning with errors, etc.) which could support stronger hardness results (although with stronger assumptions than NP != P)?

**Ethical Concerns:**

["NO or VERY MINOR ethics concerns only"]

**Final Justification:**

I recommend a score of 5 (accept).

I believe paper makes a good step towards filling the gap in the literature discussed by the authors (i.e., approximation of distributions with PCs.)

I also believe the question of probabilistic modeling using data structures / architectures that support efficient queries to be very practical and timely.

The paper is clear and well-written throughout.

**Limitations:**

yes

**Quality:**

4

**Strengths And Weaknesses:**

**Strengths**
1. I agree with the authors that prior work has studied exact representation of distributions with PCs. In fact it is surprising that more work has not been done in the present setting, where approximation is considered. This paper makes a good step to filling that gap.
2. I find the question of probabilistic modeling using data structures / architectures that support efficient queries to be well-motivated and timely.
3. I find the paper to be written in a very clear manner, even as someone who was not intimately familiar with PCs prior to reading the paper.

**Weaknesses**
1. The main weakness I find is the fact that while foundational, NP-hardness suffers practical applicability due to its own worst-case nature. In particular, this does not necessarily give hardness for even the average distribution. This limits applicability in real life situations.

---

> ### Author Rebuttal · Authors · 2025-07-31
>
> Thank you for the constructive feedback. We answer each question below.
>
> Q1. First, while NP-hardness is only concerned with worst-case complexity, we believe it is important to understand the limitations of the approximation scheme before we can move onto average cases. That being said, while outside the scope of this work, we think that this suggestion is very interesting. In fact, existing literature on expressive efficiency of different PC classes for exact representations is still limited to only worst-case complexity results. Average case complexity would be a good extension to this work, and would answer more questions about PC/tractable model representations in practice. The challenge would be in defining a good notion of ''average case'', for instance by limiting the set of distributions to be of a certain form and defining distributions over its parameter space.
>
> Q2. This is an interesting idea, we have not considered that in this work. We appreciate the suggestion and will consider this in the future.

---

> > ### Comment · Reviewer_Leaf · 2025-08-01
> >
> > Thanks for your response. Maintaining score.

---

### Official Review · Reviewer_VR5d · 2025-07-01

**Clarity:** 3
**Significance:** 2
**Originality:** 2
**Rating:** 3
**Confidence:** 3

**Summary:**

The paper investigates the ability of restricted classes of probabilistic circuits to model distributions.
3 - Approximate the distribution with a smooth decomposable PC: hard
4 - Convert decomposable PC to deterministic decomposable PC: hard
5 - If you could approximate a distribution with bounded distance, you could do approximate inference with absolute error.

**Questions:**

1. When you consider relative approximation of a distribution, how are you assuming the distribution is represented or encoded? I ask because in theorem 3.3, you start with an arbitrary formula f and define P to be the uniform distribution on models of f.
1a. Is it possible to convert a boolean formula into a PC that's uniform on models of the formula?
1b. If no: how are you encoding probability distributions as inputs?
1c. If yes: in 3.3, are you essentially showing that it's hard to convert a PC into a marginal-tractable PC?

2.  What is different between your proofs in section 4 and the proofs in Bova et al and De Colnet and Mengel? Was there any insight in the conversion to PCs?

3. Can you explain more about why hardness results for smooth decomposable PCs are significant?

**Ethical Concerns:**

["NO or VERY MINOR ethics concerns only"]

**Final Justification:**

I now agree that Theorem 3.3 is correct as stated, and the authors have clarified to a great extent the wording that confused me. I still don't understand why their choice of a model for a distribution is a good one, but I don't know enough of the background here to consider it bad either.

**Limitations:**

Yes

**Paper Formatting Concerns:**

2.1 scope definition: is the scope of n the set of leaf nodes that have a path to n, or the set of all nodes with an edge to n?
line 66: what does it mean for P to be non-negative? or rather, when can a distribution ever be negative?
3.1 you're using non-bold Y for what I think is a set of variables

**Quality:**

2

**Strengths And Weaknesses:**

+ I thought the results were well presented and organized.
+ The questions addressed by the paper seem very relevant.

? I don't know how significant the results are. The PC classes seem extremely restricted, so it doesn't surprise me that it's hard to approximate things with them, but I'm not familiar enough with the literature to say if the hardness results are interesting.
? I'm still not convinced that Boolean functions are a good way of representing a distribution, but I don't have any specific reasons for disliking them either. If I had more time I would want to continue this discussion with the authors. It's entirely possible that I'm simply missing some background that makes it logical.
? I got too stuck on theorem 3.3 to explore the rest of the paper in detail.

- Section 4 seems to be mostly a rehash of a similar proof for logical circuits

---

> ### Author Rebuttal · Authors · 2025-07-31
>
> Thank you for the constructive feedback. We answer individual questions below.
>
> Q1. When we consider relative approximation, we consider arbitrary distributions without assuming a particular representation form. For example, in Section 3.3 we consider a uniform distribution $P$ over the models of $f$, but we do not explicitly encode $P$ as a PC nor any other representation. In fact, representing this uniform distribution $P$ exactly is a non-trivial task because it would need to solve model counting which is #P-hard.
>
> Q2. The main difference between our proofs in Section 4 and those by Bova et al. [1] and De Colnet and Mengel [2] is two-fold. First, we use the Sauerhoff function, which Bova et al. used to show the exponential separation between DNNFs and d-DNNFs, to prove that the exponential gap exists even for weak approximation, defined by De Colnet and Mengel. The latter work in fact only showed that there exists a function that requires an exponential size for a d-DNNF to approximate, but not that such function can be represented by a polysized DNNF. Secondly, we extend this exponential separation to deterministic and decomposable PCs that can represent arbitrary distributions, where as previous results only applied to the case of PCs that are only allowed to represent uniform distributions over a logical formula.
>
> Q3. Understanding the expressive efficiency between different classes of PCs is of high interest to the community and well documented in the literature for the case of exact representations, but is significantly underexplored for approximations. The two PC classes that we consider are two of the most expressive efficient ones in the literature: for instance, all PC structure learning algorithms will output smooth and decomposable circuits. While the condition may seem restrictive, decomposable PCs have been shown to outperform even intractable models on density estimation tasks [3,4,5], while being tractable for marginals. Thus, finding that it is NP-Hard to approximate with decomposable PCs with a non-zero but bounded error is significant. Moreover, our NP-hardness result applies to any model that allows tractable marginal inference, and thus extends beyond decomposable PCs to a wide variety of models, such as determinantal point processes, probabilistic generating circuits, etc.
>
> Lastly, we would like to also reiterate that, to the best of our knowledge, we are the first to concretely establish relationships between approximate inference guarantees and bounded f-divergences. Previously, many types of approximate inference were implicitly assumed to hold when learning to achieve small f-divergences. This result applies generally to probabilistic models, not just PCs.
>
>
> To address your formatting concerns,
>
> - The computational graph has a scope function $\phi$ which associates each unit $n$ to a subset of the set of random variables $\mathbf{X}$, defined recursively by $\phi(n) = \cup_{c \in \text{in}(c)} \phi(c)$. The scope of a leaf node is simply the variable mentioned at that node. Alternatively, the scope of a node is the union of scopes of all leaf node with paths to that node. We will clarify this in the revision.
>
> - $P$ being non-negative is a typo, this should instead say $P$ is positive only over the models of $f$. Thank you for catching this!
>
> - $Y$ in this case is only a single auxiliary variable, not a set of variables.
>
> [1] Bova, Simone, et al. "Knowledge Compilation Meets Communication Complexity." IJCAI. Vol. 16. 2016.
>
> [2] De Colnet, Alexis, and Stefan Mengel. "Lower bounds for approximate knowledge compilation." Proceedings of the Twenty-Ninth International Conference on International Joint Conferences on Artificial Intelligence. 2021.
>
> [3] Gala, Gennaro, et al. "Probabilistic integral circuits." International Conference on Artificial Intelligence and Statistics. PMLR, 2024.
>
> [4] Liu, Anji, and Guy Van den Broeck. "Tractable regularization of probabilistic circuits." Advances in Neural Information Processing Systems 34 (2021): 3558-3570.
>
> [5] Liu, Anji, Honghua Zhang, and Guy Van den Broeck. "Scaling Up Probabilistic Circuits by Latent Variable Distillation." The Eleventh International Conference on Learning Representations.

---

> > ### Comment · Reviewer_VR5d · 2025-08-01
> >
> > The input is "a probability distribution $P$ and a $k$-convex $f$-divergence $D_f$".
> >
> > The reduction proceeds by converting a boolean formula $\phi$ to $\phi'$ which has one additional satisfying assignment, defining a uniform distribution $P$ on satisfying assignments of $\phi'$, and showing that if $Q$ is a $k\epsilon^2$-approximation wrt $D_f$, then there is a marginal distribution on $Q$ that is $\geq 1/4$ iff $\phi$ is satisfiable. This much seems correct to me.
> >
> > The claim is: this implies that given a distribution $P$, it's NP-hard to compute a $k\epsilon^2$ $D_f$-approximation of $P$ because doing so would solve SAT. However, this doesn't necessarily follow.
> >
> > Problem 1: suppose we have an algorithm to compute a $D_f$-approximation of an arbitrary distribution $P$ in polynomial time. In order to solve SAT using this, we would need to first construct a uniform distribution on the satisfying assignments of $\phi$, which as you say is #P-hard.
> >
> > Problem 2: what does it mean to compute such an approximation in polynomial time, when the input is an arbitrary probability distribution independent of representation? The time bound is bounded by a polynomial of what variable?

---

> > > ### Author Response · Authors · 2025-08-02
> > >
> > > Thank you for clarifying the issue with the proof of Theorem 3.3. We will correct the theorem statement as following to be more precise:
> > > ''Given a (potentially unnormalized) probability distribution $\hat{P}$ and a $k$-convex f-divergence $D_f$, for any $0 < \epsilon < 1/4$, it is NP-hard to represent the $k\epsilon^2$-$D_f$-approximation of its normalized distribution $P$ as a model that can tractably compute marginals''
> > >
> > > The proof statement would remain largely unchanged, except for the input being an unnormalized distribution $\hat{P} = f'$. The theorem would still be of interest given that many probabilistic models that we wish to compile and perform inference represent unnormalized distributions: e.g., factor graphs, energy-based models, Markov logic networks, and many probabilistic (logic) programs such as ProbLog. Thus, our hardness result covers many forms of ''ground-truth distributions'' and still applies to any model that can tractably marginalize as the approximator.
> > >
> > > To answer your second question, we are concerned with the approximation being in polynomial time with respect to the size of the model representing $\hat{P}$, using the appropriate notion of size for each particular model class. This is in line with how we define polynomial time compilation of a model: e.g. if the input is a CNF, then polynomial in the number of variables and clauses; if the input is a graphical model, then polynomial in the number of nodes (variables) and edges. In particular, for the proof of Theorem 3.3, if we can approximate $P$ using a PC in polynomial time w.r.t. the size of $\hat{P}$ (i.e. $f'$), then we can answer SAT by marginalizing the circuit in time polynomial in the size of $f'$ which in turn is polynomial in the size of $f$.

---

> > > > ### Comment · Reviewer_VR5d · 2025-08-02
> > > >
> > > > Thank you for the quick response! I'm not familiar with unnormalized distributions, though - how do you construct a model of an unnormalized distribution of satisfying assignments to $f$ in polynomial time?

---

> > > > > ### Author Response · Authors · 2025-08-02
> > > > >
> > > > > The unnormalized distribution is $f$ itself, and its normalized counterpart will return $1/MC(f)$ for satisfying assignments of $f$ and 0 otherwise. Here $MC(f)$ denotes the model count of $f$.

---

> > > > > > ### Comment · Reviewer_VR5d · 2025-08-02
> > > > > >
> > > > > > Sorry, I still don't see how to convert a Boolean formula to a model of a distribution that's uniform on solutions to the formula.
> > > > > > First, I want to clarify: when you say the unnormalized distribution is $f$, are you considering the Boolean formula as a model of the distribution? Or are you modeling the distribution with a probabilistic circuit in some way and identifying the model with $f$?
> > > > > > I can see how to construct a PC that models a distribution whose support is solutions, but everything I've tried has been non-uniform on those solutions.

---

> ### Author Response · Authors · 2025-08-02
>
> We do not need to explicitly construct any model that represents a uniform distribution over the satisfying assignments. The problem that we are showing is NP-hard is to take as an input an unnormalized distribution $\hat{P}$, which can really be any non-negative function, and output a PC representing a distribution that is close (according to $D_f$) to the normalized distribution $P$ defined as: $P(\mathbf{x}) = \hat{P}(\mathbf{x})/Z$ where $Z=\sum_\mathbf{x} \hat{P}(\mathbf{x})$ is the normalizing constant. We know that $\hat{P} \propto P$, and an algorithm to solve this problem, hypothetically, could output the approximation without ever constructing $P$ explicitly.
>
> In our proof, we reduce the SAT problem for formula $f$ by constructing $f'$ from it as described in the paper, and give it as input to our ``approximate modeling' problem. Here, $f'$ can be seen as a Boolean function where $f'(\mathbf{x})=1$ if $\mathbf{x}$ is a satisfying assignment and $f'(\mathbf{x})=0$ otherwise; because it is non-negative, we can treat it as an unnormalized distribution. The goal is to output a PC representing a distribution close to $P = f'(\mathbf{x})/MC(f')$ which we show is NP-hard because such PC can be used to answer SAT for the original formula $f$. We hope this answers your question.

---

> > ### Comment · Reviewer_VR5d · 2025-08-02
> >
> > Yes, that answers my question. Thank you!
> > It seems to me then that the model-independence of the theorem is actually a weakening of the result? That is, 3.3 doesn't rule out, for example, converting a PC P into a $D_f$-approximate marginal-tractable PC $Q$. It only shows that there is no polynomial-time algorithm to convert an arbitrarily-specified model of P into a marginal-tractable PC $Q$. Is that correct?

---

> > > ### Author Response · Authors · 2025-08-02
> > >
> > > Great observation! Indeed, this does not rule out polynomial-time approximation of a **normalized** distribution using decomposable PCs.
> > >
> > > A subtle point to note here is that ensuring normalization is not as trivial as locally normalizing each sum node (i.e. edge parameters sum to one) in the case of PCs that are not marginal-tractable. A simple example circuit of this is: $(0.5 (X) + 0.5 (\neg X)) \times (X)$. While the circuit is locally normalized, it returns probability 0.5 for X=true and 0 for X=false; thus the distribution is still unnormalized.

---

> > > > ### Comment · Reviewer_VR5d · 2025-08-03
> > > >
> > > > Let me try explaining my issue again.
> > > > Theorem 3.3 doesn't say anything about the difficulty of computing marginals. It says that it's hard to extract information from an arbitrary model. Consider the following strengthening:
> > > >
> > > > Theorem 3.3+: Let $P$ be an arbitrary probability distribution. Then computing a $k\epsilon^2$-$D_f$-approximation in a model with tractable marginals is uncomputable.
> > > > Proof: Let $M$ be an arbitrary Turing machine. Construct $M'$ which accepts $\epsilon$ and {$0x\mid x\in L(M)$}. Consider $M'$ to represent a uniform distribution on $L(M')$. Suppose we could construct a distribution $Q$ that approximates $M'$ in a model that could tractably compute marginals. Then if $Q(\epsilon)= 1$,  $L(M)=\emptyset$ and if $Q(\epsilon)\leq \frac{1}{2}$, $L(M)\not=\emptyset$. But determining if $L(M)=\emptyset$ is undecidable.
> > > >
> > > > Even if you consider this model silly, there's problems with using Boolean formulas as models of distributions:
> > > >
> > > > Theorem 3.3*: Let $P$ be an arbitrary probability distribution. Then computing a $k\epsilon^2$-$D_f$-approximation in a model that can sample is NP-hard.
> > > > Proof: As in Theorem 3.3, but sample $Q$ and estimate $\Pr[Y=1]$.
> > > >
> > > > But no one would say "sampling is NP-hard". The difficulty is in the implicit "given a completely arbitrary model of a distribution, get any information out of it".
> > > >
> > > > As a result, the claim in lines 209 and 210 doesn't follow. The class of poly-sized smooth and decomposable PCs could contain an approximation of every distribution; what's hard is converting a Boolean formula to a PC that's uniform on solutions to that Boolean formula.

---

> > > > > ### Author Response · Authors · 2025-08-03
> > > > >
> > > > > > But no one would say "sampling is NP-hard".
> > > > >
> > > > > On the contrary, sampling solutions of a combinatorial problem is at least as hard as showing the existence of a solution [1]. That is, uniformly sampling satisfying assignments of a Boolean formula (and returning empty if no such assignment exists) would answer SAT and is NP-hard. In fact, even *almost-uniform sampling*--which is what your suggested proof for Theorem 3.3* would do--is inter-reducible with approximate counting [1] and would be hard wherever approximate counting is hard: e.g., approximate model counting is #P-hard [2].
> > > > >
> > > > > > The class of poly-sized smooth and decomposable PCs could contain an approximation of every distribution.
> > > > >
> > > > > Our proof of Theorem 3.3 already provides a class of distributions that is a counterexample to the above statement: namely, $f'$ constructed from a Boolean formula $f$ that is in a class of representations for which SAT is NP-hard. Whether or not we can compute/represent the uniform distribution over models of $f'$ in polynomial time, the distribution certainly exists and is well-defined ($P(\mathbf{x})=1/MC(f')$ if $\mathbf{x} \models f'$, P(\mathbf{x})=0 otherwise). Our proof shows that unless P=NP, such distribution cannot be contained in the set of distributions that can be *exactly or approximately* modeled by polysized smooth and decomposable PCs.
> > > > >
> > > > > [1] Jerrum, M.R., Valiant, L.G. and Vazirani, V.V. "Random generation of combinatorial structures from a uniform distribution". Theoretical computer science. 1986.
> > > > > [2] Chakraborty, S., Meel, K.S., and Vardi, M.Y. "Approximate model counting." Handbook of Satisfiability. 2021.

---

> > > > > > ### Author Response · Authors · 2025-08-06
> > > > > >
> > > > > > Thank you for engaging in the discussion of our work. We wanted to follow up to see if everything is clear now, or if you have any further questions. Please let us know if there is anything else we can help with.

---

> ### Comment · Reviewer_VR5d · 2025-08-08
>
> Sorry, got busy for a few days.
> I agree, sampling solutions to a combinatorial problem is at least as hard as determining existence of a solution. My point is that the difficulty lies in starting with an instance of a combinatorial problem, and that the difficulty of sampling depends on the representation of the distribution. I don't think my example made that clear, though.
>
> "Our proof shows that unless P=NP, such distribution cannot be contained in the set of distributions that can be exactly or approximately modeled by polysized smooth and decomposable PCs."
>
> I still don't think this is true- what your proof shows is that you can't convert a boolean function $f$ into a poly-sized smooth decomposable PC that is uniform on solutions to $f$ in polynomial time, unless P=NP. That's different from saying that you can't have a poly-sized smooth decomposable PC that is uniform on solutions to $f$. In fact, for any poly-sized smooth decomposable PC that models a uniform distribution, there are boolean functions whose solution set is the support of $P$. You just can't compute the PC from the function (in polytime unless P=NP).
>
> Going back to your original theorem statement, this is why I'm so hung up on how you represent the distribution $P$. If $P$ is represented as a boolean formula, then the problem is at least as hard as solving SAT. I agree that Theorem 3.3 proves this. I believe my Theorem 3.3+ is also correct: for some representations of $P$, the problem is uncomputable.
>
> So I think what's going on is that
> 1) I had different expectations about what the theorem was proving. I initially assumed you were proving that it's hard to convert an arbitrary PC $P$ into a poly-sized marginal-tractable approximation of $P$.
>
> I think my expectations were wrong because when you discuss models of distributions you primarily discuss PCs, so I never considered a boolean function to be a model of a distribution.
>
> 2) there are some default assumptions about what a valid model of a distribution is, which I'm not aware of. (Solution to boolean function: good, halting input for Turing machine: silly.)
>
> I personally would like you to make explicit your assumptions about what a valid model of a distribution are. I believe this would fix both issues 1 and 2 for me simultaneously.
>
> Additional notation comment: you use $f$ for both the function in $D_f$ and the boolean function. I think you should fix this one, it's not likely to confuse but it's easy to fix.
>
> Thank you for your patience in reading all of these!

---

> > ### Author Response · Authors · 2025-08-08
> >
> > > what your proof shows is that you can't convert a boolean function $f$ into a poly-sized smooth decomposable PC that is uniform on solutions to $f$ in polynomial time, unless P=NP. That's different from saying that you can't have a poly-sized smooth decomposable PC that is uniform on solutions to $f$.
> >
> > Apologies, yes, this is indeed what our theorem proves. The goal was to extend the hardness of exact compilation into the approximate case: it is not only NP-hard to exactly compile a (potentially unnormalized) distribution into a decomposable PC but also to do so approximately with a small $\epsilon$.
> >
> > > there are some default assumptions about what a valid model of a distribution is, which I'm not aware of. (Solution to boolean function: good, halting input for Turing machine: silly.)
> >
> > Our updated theorem statement assumes a (potentially unnormalized) probability distribution $\hat{P}$ over some set of variables $\mathbf{X}$. An implicit assumption throughout the paper is that $\mathbf{X}$ is a finite set. Any (unnormalized) distribution $\hat{P}: \textsf{val}(\mathbf{X}) \to \mathbb{R}$ must satisfy the following: (i) $\hat{P}(\mathbf{x}) \geq 0$ for any $\mathbf{x}$, and (ii) the normalizing constant $Z = \sum_{\mathbf{x} \in \textsf{val}(\mathbf{X})} \hat{P}(\mathbf{x})$ is well-defined and finite. A Boolean function satisfies this requirement, but a Turing machine does not. More generally, a language of a Turing machine can be countably infinite, and it is well known that a uniform distribution over a countably infinite set cannot exist. We will make these assumptions explicit in the revision. Hopefully this addresses both issues 1 and 2 you've raised.
> >
> > > Additional notation comment: you use $f$ for both the function in $D_f$ and the boolean function. I think you should fix this one, it's not likely to confuse but it's easy to fix.
> >
> > Good point! We will fix this in the revision to avoid confusion; e.g., we can use $g$ for the Boolean function.
> >
> > Thank you again for engaging in the discussion, and we hope that we have addressed your concerns. Just a reminder, the reviewer-author discussion period will be closed in less than 24 hours.

---

> > > ### Comment · Reviewer_VR5d · 2025-08-08
> > >
> > > Can't you still push the hardness above NP by choosing the distribution model to be instances of, say, a PSPACE-complete problem representing a uniform distribution on solutions to that instance?
> > > I think even the Turing machine example is still valid: choose only Turing machines that halt on a finite set of inputs, representing a uniform distribution on that finite set. You can't require a condition like "we can efficiently determine if the input is a valid distribution", because an unsatisfiable Boolean formula doesn't represent a valid distribution either.

---

> > > > ### Author Response · Authors · 2025-08-08
> > > >
> > > > We believe a Turing machine based argument could potentially work to show a much stronger result of undecidability, and will add this to the paper following the construction of a formal argument. However, we struggle to see how this invalidates our claims pertaining to Theorem 3.3 as we have shown it is NP-hard, though it may be stronger if the Turing machine argument holds.
> > > > To summarize the discussion,
> > > > - We will correct Theorem 3.3 to be more precise w.r.t. the input being potentially unnormalized, and clarify our assumptions about valid probability distributions.
> > > > - We will add a more in-depth definition of the scope of a node in a PC.
> > > > - Correct the typo of $P$ being non-negative
> > > > - Change the Boolean formula to $g$ instead of $f$ to prevent confusion with $D_f$
> > > >
> > > > As the discussion phase is closing soon, we would appreciate it if the reviewer would consider updating their score, or provide any other questions we should address.
> > > >
> > > > >You can't require a condition like "we can efficiently determine if the input is a valid distribution", because an unsatisfiable Boolean formula doesn't represent a valid distribution either.
> > > > >
> > > > We do not claim that we can efficiently determine if the input is a valid distribution, but we do assume this for the inputs to our approximator. This is why we construct $f’$ such that it forms a valid distribution whether or not the original Boolean formula is satisfiable.

---

> > > > > ### Author Response · Authors · 2025-08-08
> > > > >
> > > > > Upon further consideration, we believe we have a better understanding of the reviewer's concerns. While approximating a uniform distribution over a halting set of a Turing machine would solve the halting problem, the difficulty here lies in the fact that the unnormalized distribution $\hat{P}$ itself cannot be evaluated. Instead, our theorem states that even for distributions $\hat{P}$ that can be evaluated efficiently, it is NP-hard to approximate them closely.

---

> > > > > > ### Comment · Reviewer_VR5d · 2025-08-08
> > > > > >
> > > > > > I'm not sure what you mean by "evaluating" a distribution, but as long as you can clarify what the conditions are for something to be a good model for a distribution, I'm happy. I'll update my score as soon as possible. Thank you again for your patience!

---

> > > > > > > ### Author Response · Authors · 2025-08-09
> > > > > > >
> > > > > > > What we mean by “evaluating” is that the value of $\hat{P}(\mathbf{x})$ can be computed efficiently for any $\mathbf{x}$. This is not the case for a distribution over the halting set, but can be done for distributions over models of CNFs and other probabilistic models.
> > > > > > >
> > > > > > > Thank you for taking the time to review our work! We appreciated the discussion and will be sure to add the clarifications in revision.

---

### Note · Authors · 2025-08-12

We thank the reviewers for the helpful discussion and summarize it in the following:

**Theorem 3.3:**
- We adjust Theorem 3.3 to be more precise: “Given **a (potentially unnormalized) probability distribution $\hat{P}$** and a $k$-convex f-divergence $D_f$, for any $0<\epsilon< 1/4 $ it is NP-hard to represent the $k\epsilon^2$-$D_f$-approximation of its normalized distribution $P$ as a model that can tractably compute marginals.”
- We will also clarify what constitutes a valid distribution. Any (unnormalized) distribution $\hat{P}: val(\mathbf{X}) \to \mathbb{R}$ must satisfy the following: (i) $\hat{P}(\mathbf{x}) \geq 0$ for any $\mathbf{x}$; and (ii) the normalization constant $Z = \sum_{\mathbf{x} \in val(\mathbf{X})} \hat{P}(\mathbf{x})$ is well-defined and finite. Moreover, throughout the paper we assume that $\mathbf{X}$ is a finite set.
- The adjusted Theorem 3.3 (proof relying on an unnormalized distribution) is still of interest to the community as many probabilistic models represent unnormalized distributions, such as factor graphs, energy based models, etc. We also point out that distributions over Boolean functions have been well studied, as they are closely related to the (weighted) model counting problem, and especially in the context of succinctness of circuit classes, although again primarily for exact representation [Choi and Darwiche, 2017; Choi et al., 2020].

**Clarify contributions:**
- Upon closer inspection, De Colnet and Mengel (2021) did not show the exponential separation between DNNF and d-DNNF persists under approximation. We will revise to highlight that this is also a new result.
- We will also highlight that our results (outside of section 4) apply more generally to (tractable) probabilistic models and inference, thus applying to a broader audience.
- We will add a discussion on related works on approximate (weighted) model counting.

**Future work:** While this is the first work on studying hardness of approximations with probabilistic circuits, as reviewers have pointed out, the paper mostly contains negative results. We hope that this paper will motivate other works on approximations with inference guarantees as well as further studies in PC classes for approximate representations. We thank the reviewers for suggesting interesting directions such as probabilistic approximations and average-case guarantees.

---

### Decision · Program_Chairs · 2025-09-17

**Decision:**

Accept (spotlight)

**Comment:**

This work presents fundamental theoretical results on the hardness of approximating distributions with probabilistic circuits. The submission received strong support with three reviewers giving strong accepts and one borderline reject. During extensive exchange between the authors and Reviewer VR5d, the technical concerns were thoroughly discussed, leading to clarifications and a better understanding of the results. Reviewer VR5d also acknowledged the technical correctness of the proofs in the final justification of the review.

Overall, the paper makes significant contributions by establishing the first systematic study of approximation hardness for probabilistic circuits, proving NP-hardness for bounded f-divergence approximations and exponential separations between circuit classes. These theoretical foundations are crucial for understanding the fundamental limits of tractable probabilistic modeling. Given the overwhelming reviewer consensus, technical soundness, and importance of the contributions to the probabilistic modeling community, I recommend **ACCEPT**.